# MINT: Minimal Information Neuro-Symbolic Tree for Objective-Driven Knowledge-Gap Reasoning and Active Human Elicitation

## Abstract

Joint planning through language-based interactions is a key area of human-AI teaming. Planning problems in the open world often involve various aspects of incomplete information and unknowns, e.g., objects involved, human goals/intents, and their impacts on planning – thus leading to knowledge gaps in joint planning. In this paper, we consider the problem of discovering optimal interaction strategies for AI agents to actively elicit human inputs in object-driven planning. To this end, we propose Minimal Information Neuro-Symbolic Tree (MINT) to reason about the impact of knowledge gaps and leverage self-play with MINT to optimize the AI agent's elicitation strategies and queries. More precisely, MINT builds a symbolic tree by making propositions of possible human-AI interactions and by consulting a neural planning policy to estimate the uncertainty in planning outcomes caused by remaining knowledge gaps. Finally, we leverage LLM to search and summarize MINT's reasoning process and curate a set of queries to optimally elicit human inputs for best planning performance. By considering a family of extended Markov decision processes with knowledge gaps, we analyze the return guarantee for a given MINT with active human elicitation. Our evaluation on three benchmarks involving unseen/unknown objects of increasing realism shows that MINT-based planning attains near-expert returns by issuing a limited number of questions per task while achieving significantly improved rewards and success rates.

## 1 Introduction

The ability for AI agents and humans to jointly plan and solve problems in the open world is instrumental to human-AI teaming. It is expected that AI agents will function as teammates instead of simply a tool amplifying human capabilities or generating human-like responses (Xu & Gao, 2023). To this end, Large Language Models (LLMs) (Xiao & Wang, 2023) and Agentic AI (Carroll et al., 2019) have demonstrated their potential in joint planning with humans (e.g., navigation, task support, and collaboration (Zu et al., 2024; Zhang et al., 2024)) through natural-language interactions. However, joint planning in the open world is rarely about solving a well-defined or known problem – various aspects of the planning problem, such as objects involved, human goals/intents, and their impact on planning (e.g., with respect to transition, constraints, and rewards), are often not completely known to agents, resulting in potential knowledge gaps. Ignoring or planning through the uncertainty may lead to conservative executions and non-optimal plans (Lockwood & Si, 2022).

The problem can be addressed by AI agents actively eliciting necessary human inputs (in the form of language-based interactions) toward bridging knowledge gaps and supporting task objectives. However, existing solutions such as Reinforcement Learning with Human Feedback (RLHF) and few-shot adaptation in LLMs (Bai et al., 2022; Ji et al., 2023) are able to process additional human feedback, but lack the ability to expressively reason and actively elicit human inputs with respect to planning objectives (Huang et al., 2022). On the other hand, Human-in-the-Loop (HITL) approaches (Mandel et al., 2017) and interactive learning (Teso & Kersting, 2019) allow fine-tuning using recommendations or expert demonstrations (Mosqueira-Rey et al., 2023). But they do not directly address the knowledge gaps nor support objective-driven active elicitation.

We propose Minimal Information Neuro-Symbolic Tree (MINT) to reason about the impact of knowledge gaps and leverage self-play with MINT to optimize the AI agent's elicitation strategy of human inputs in objective-driven planning. More precisely, MINT defines a symbolic tree, where each node represents a state of the planning problem with a corresponding knowledge gap, while each edge represents a proposition of possible human-AI interaction to narrow down the knowledge gap in child nodes. We build and expand MINT by consulting a neural planning policy, more specifically, in this work we use UA-DQN, an uncertainty-aware DQN algorithm based on bootstrapped distributional RL (Clements et al., 2019) to estimate the uncertainty in planning outcomes caused by the remaining knowledge gap of each child node, until each node has a unitary impact on planning outcomes. Thus, we enable a self-play process using MINT to analyze how different interactions between human and AI can bridge the knowledge gap and lead to optimized planning results. Finally, we leverage LLM to search and summarize MINT's reasoning process and curate a set of questions/queries to optimally elicit human inputs for best planning performance.

By considering a family of extended Markov Decision Processes (MDPs) with knowledge gaps, we analyze the impact of the final knowledge gap in a given MINT and for a set of identified questions (from AI to human). We prove a local pseudo-Lipschitz continuity of the planning returns and provide an upper bound on the return-gap between MINT-based planning and an ideal case without any knowledge gap. The results provide a performance guarantee for our MINT-based planning with active human elicitation. To the best of our knowledge, this is the first work combining symbolic reasoning of knowledge gaps, neural planning policy, and LLM to enable active and optimized human elicitation in language-based planning. We evaluate MINT on three benchmarks involving unseen/unknown objects (affecting transitions or rewards) of increasing realism: MiniGrid; Atari ALE – modified Atari games, and real-world search and rescue in Isaac Gym. Across all domains, MINT-based planning attains near-expert returns by issuing only 1–3 binary questions per object, while achieving a significantly increased reward and success rate over baselines.

The primary contributions of this paper are threefold:

- We propose MINT, which consults a neural planning policy and symbolically reasons about the impact of knowledge gaps in planning. Through self-play with relevant propositions, it enables active and optimized human elicitation using LLM toward planning objectives.

- We show local pseudo-Lipschitz continuity of the planning returns and provide an upper bound on the return-gap between MINT-based planning and an ideal case without any knowledge gap, providing solid support to MINT-based planning.

- Through empirical evaluations, we demonstrate that MINT-based planning attains near-expert returns with almost an order of magnitude fewer questions (from AI to human), in challenging RL environments such as Minigrid, Atari Pacman and Isaac drone environment.

## 2 RELATED WORKS

**Language-Based Planning**    Language-based planning has been researched before the rise of LLMs. Previous methods integrate the planning into the RL process regarding query generation as part of the action space. The agent will learn to execute language query commands when having difficulty in decision making (Liu et al., 2022; Nguyen et al., 2021). However, these methods could only generate simple queries with predefined vocabulary used. Recently, advances in LLMs show great potential for solving language-based joint planning tasks with human-AI cooperation. Novel frameworks have been proposed combining LLMs with classical AI planning agents in a variety of fields including robotic and embodied AI (Izquierdo-Badiola et al., 2024; Dai et al., 2024), games (Wu et al., 2023b; , FAIR), creative writing (Chakrabarty et al., 2022) and real-world task planning (Xie et al., 2024). However, these approaches have a limited ability in understanding and reasoning about knowledge gaps and the resulting impact, which requires a better elicitation strategy of human knowledge.

**Planning under Partial Observability.**    Partial observability poses a major challenge for classical RL that assumes the Markov property. Recent research tackles this problem in two broad ways: (1) Ignoring or minimizing the knowledge gap, *i.e.*, training agents that do not explicitly model unseen information or perform conservative and safe planning (Meng et al., 2021; Ni et al., 2021) (2) Incorporating robust planning or Bayesian-type belief-state reasoning by maintaining an internal representation of uncertainty or plan over a range of possible states, such as latent belief state (Wang

et al., 2023), Bayesian inference (Gu et al., 2021), or ensembles (Ghosh et al., 2021). These approaches do not apply to any knowledge gaps (e.g., unseen objects or unknown goals) and lack the ability to interact with humans in joint planning problems in order to elicit inputs and bridge the knowledge gap.

**Human-in-the-Loop RL**  Existing works on HITL RL often focus on the phase of agent learning, in which the human provides direct evaluations or preferences on agents' actions as feedback (Christiano et al., 2017; Rafailov et al., 2024). Another underexplored paradigm is the incorporation of human demonstrations or advice on actions of expertise (Luo et al., 2023; Wu et al., 2023a; Igbinedion & Karaman, 2024), which requires a domain-expertise policy. Recent methods further reduce the communication burden by only asking humans when facing high uncertainty in decision-making (Mandel et al., 2017; Da Silva et al., 2020; Singi et al., 2024). However, expertise actions are not always available. Instead, the agent needs to analyze the knowledge gap first, then precisely query humans for the key information missing, and finally conclude optimal actions by itself.

**Neuro-Symbolic RL**  Neuro-symbolic RL (NS-RL) aims to combine the representational power of neural networks with the interpretability offered by symbolic methods. Compared with classical interpretable RL approaches (e.g., post-hoc saliency (Greydanus et al., 2018) or decision tree distillation (Chen et al., 2024)), NS-RL allows for more explicit knowledge integration and rule-based reasoning, thereby improving both interpretability and generalizability. Recent works have proposed diversified approaches for NS-RL implementation, such as DiffSES (Zheng et al., 2024) and INSIGHT (Luo et al., 2024). Besides, Jin et al. (2022) and Lyu et al. (2019) integrate symbolic planning and options as a perception with deep RL to solve tasks with high-dimensional inputs. Other works consider integrating symbolic methods into policy representation, including deep symbolic policy search (Landajuela et al., 2021) and programmatic RL (Verma et al., 2018; 2019).

## 3 PRELIMINARIES

Planning problems are often formulated as Markov Decision Processes (MDPs). For human-AI joint planning in the open world, various aspects of the planning problem, such as objects, environment factors, and human intents, may not be known to the AI agents, leading to uncertainties in decision-making. While there may be many different types of unknowns in the open world, they eventually impact the MDP's state transition probability and/or the reward function. Hence, we collectively define such unknowns as a knowledge gap $u$. We model this planning problem by an *MDP family with knowledge gap*, denoted by $\mathcal{M} = (\mathcal{S}, \mathcal{A}, u, \mathcal{T}_u, \mathcal{R}_u, \gamma)$, where $\mathcal{S}$ and $\mathcal{A}$ are the state and action spaces, and $\gamma$ is a discount factor, similar to standard MDPs. However, due to the knowledge gap $u$, we may not know the exact state transition probability and reward function. We model such uncertainty through two sets, $\mathcal{T}_u$ and $\mathcal{R}_u$ respectively, given the knowledge gap $u$. Thus, $T(s'|s, a) \in \mathcal{T}_u : \mathcal{S} \times \mathcal{S} \times \mathcal{A} \to [0, 1]$ is a possible probability distribution of state transition, and $R(s, a) \in \mathcal{R}_u : \mathcal{S} \times \mathcal{A} \to \mathbb{R}$ is a possible reward function, under the knowledge gap $u$.

We consider the problem of actively eliciting human inputs to bridge the knowledge gap and to optimize planning objects. To this end, we consider a parametric representation of the knowledge gap and uncertainty sets. Let $\varphi$ be a descriptor vector uniquely defining the transition probability distribution $T_\varphi(s'|s, a)$ and the reward function $R_\varphi(s, a)$. We can rewrite the uncertainty set using $\Phi_u \subseteq \mathbb{R}^d$, where $d \in \mathbb{N}$ is fixed and $\Phi_u$ is the descriptor space under knowledge gap $u$. For each descriptor $\varphi \in \Phi_u$, the corresponding MDP, $M_\varphi = (\mathcal{S}, \mathcal{A}, T_\varphi, R_\varphi, \gamma)$ becomes a standard known one. The extended MDP family under knowledge gap $u$ is therefore defined as $\mathcal{M}_u = \{M_\varphi \mid \varphi \in \Phi_u\}$. For a given descriptor $\varphi \in \Phi_u$ (thus zero knowledge gap), the corresponding MDP, $M_\varphi$ can be solved using standard approaches like RL (Mnih et al., 2015). This is achieved by finding a planning policy to maximize the expected discounted cumulative rewards, denoted as the return $J$:

$$J(\pi|\varphi) \; = \; \mathbb{E}\Big[\sum_{t=0}^\infty \gamma^t\, R_\varphi(s_t, a_t)\Big], \quad \text{where } (s_0, a_0, s_1, \dots) \sim M_\varphi, \pi. \tag{1}$$

Existing RL algorithms can be leveraged to learn an optimal policy $\pi_\varphi^*(a|s)$ to maximize the return, i.e., $\pi_\varphi^* = \arg\max_\pi J(\pi|\varphi)$. In particular, in the value-based deep RL with discrete action spaces, we use neural networks to fit the optimal action value-function, known as Q-function, which estimates the return under the current state and action: $Q_\varphi^*(s, a) = \max_\pi J(\pi|\varphi; s_0 = s, a_0 = a)$. Thus, the

optimal policy can be determined by taking the optimal action $a_t^* = \pi^*(s) = \arg\max_a Q^*(s_t, a)$ when no knowledge gap exists. In this paper, we focus on eliciting human inputs via natural language to narrow down the knowledge gap $\Phi_u$ and thus achieve optimal planning performance.

# 4  OUR PROPOSED SOLUTION USING MINT

Consider an MDP in state $s$ and with initial knowledge gap $u_0$. To reason about the impact of $u_0$, we leverage a symbolic tree representation in MINT and expand it by consulting a neural planning policy. More precisely, starting with a root node representing $u_0$, we consider a sequence of propositions regarding possible human interactions, in the form of potential queries denoted by $q_k = \xi(u_k, s)$ at each step $k$, to elicit a corresponding human response $y_k$ for each query. In this paper, we define each query as a yes/no proposition using natural language, and thus each response is a binary answer $y_k \in \{0, 1\}$ based on the human's knowledge of the latent ground-truth.

By considering such propositions in a self-play process, we build MINT as a symbolic tree representation with the knowledge gap as nodes and propositions as edges. Thus, $(q_k, y_k)$ at each step $k$ iteratively narrows down the knowledge gap in resulting child nodes (depending on different possible human responses), i.e., $\Phi_{u_k}$ with $u_{k+1} = F_{q_k, y_k}(u_k)$ by a processing function $F$. It leads to reduced uncertainty in the potential descriptor space with $\Phi_{u_{k+1}} \subset \Phi_{u_k}$ at each knowledge update step $k$. Let $\varphi^*$ be the latent ground-truth descriptor and thus $J(\pi_{\varphi^*}^* | \varphi^*)$ the ideal return without knowledge gap. We aim to find an optimal interaction/query strategy $\xi$ (with complexity $K$) for generating queries and eliciting human responses, so that the return gap at the terminal knowledge gap $u_K$ is minimized:

$$\xi^* = \arg\min_\xi \left[ \max_{\varphi \in \Phi_{u_K}} J(\pi_{\varphi^*}^* | \varphi^*) - J(\pi_\varphi^* | \varphi) \right], \text{ s.t. } u_{k+1} = F_{q_k, y_k}(u_k), \ q_k = \xi(u_k, s), \forall k. \quad (2)$$

Figure 1 shows the key steps in MINT construction and its execution. The first step is to train a neural planning policy (i.e., bootstrapped DQN) that estimates not only $Q_\varphi^*(s, a), \forall \varphi$, but also the variance of the optimal Q-value $\sigma_u^2(s) = var_{\varphi \sim \Phi_u}(Q_\varphi^*(s, a^*))$ to quantify the impact of knowledge gap $u$ on planning outcomes. Next, when the AI agent detects the existence of a knowledge gap $u_0$ (i.e., objects with low confidence or ambiguous human intent), it expands MINT from $u_0$ by considering propositions that further split the current gap into child nodes (in a self-play process) and evaluating their impact on planning through $\sigma_u^2(s)$, until reaching the depth limit or the variance/impacts are small enough. Finally, an LLM will process the resulting MINT by merging equivalent sub-trees or leaves with identical optimal actions. It then curates a binary query $q$ dividing the tree to maximize the information gain relating to the optimal action candidates in leaf nodes, given the current tree structure and prompts. Each query is presented to the human in joint planning to get a response $y$, which is utilized to reduce the knowledge gap during execution. This query-and-answer process iteratively interacts with the human to minimize the return gap in objective-oriented joint planning.

## 4.1  EVALUATING THE IMPACT OF CURRENT KNOWLEDGE GAPS

The solution in this work consists of two phases: training and deployment. As defined in Section 3, during the training process, we consider the extended MDP family $\mathcal{M}_u$ with a knowledge gap $u$. Given a specific $\varphi \in \Phi_u$ in the knowledge gap known to the agent, the optimal Q-function $Q_\varphi^*(s, a)$ conditioned on $\varphi$ can then be estimated using regular RL approaches. Besides, to estimate the impacts of a given knowledge gap $u$, we train a bootstrapped DQN to estimate the variance of optimal Q-values over $\varphi \sim \Phi_u$ for any $u$ and $(s, a)$, as described later in the training session.

At deployment, the ground-truth descriptor $\varphi^*$ is unknown. The AI agent maintains a knowledge vector $u_k \in \mathbb{R}^d$, which is associated with $\Phi_{u_k}$, s.t. $\varphi^* \in \Phi_{u_k} \subset \Phi_u$, limiting the possible range of $\varphi^*$. Initially, we have $u = u_0$. In each timestep $t$, the agent can choose to generate a proposition with query $q_k$ in natural language and self-play potential response $y_i \in \{0, 1\}$. Then, the current knowledge $u_k$ will be updated to $u_{k+1}$ based on $(q_k, y_k)$, iteratively narrowing down the knowledge gap: $\varphi^* \in \Phi_{u_{k+1}} \subset \Phi_{u_k} \subset \cdots \subset \Phi_u$. At the end of this iteration, we aim to bound the return gap within a reasonable value $\varepsilon$ at the terminal knowledge gap $u_K$ as : $\max_{\varphi \in \Phi_{u_K}} \|J(\pi_{\varphi^*}^* | \varphi^*) - J(\pi_\varphi^* | \varphi)\| \le \varepsilon$. We also provide theoretical analysis to demonstrate that this return gap can be bounded by the range of $\Phi_{u_k}$, thus justifying our approaches of iteratively narrowing down $\Phi_u$ via queries.

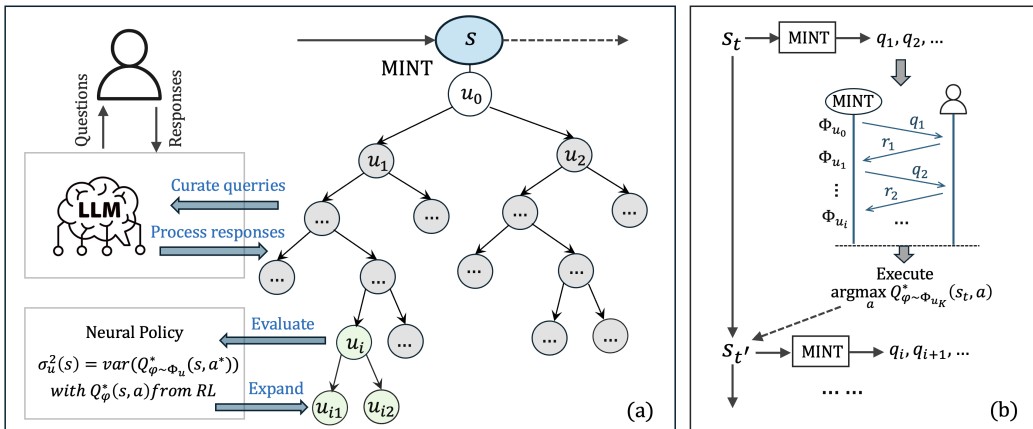

Figure 1: **Evaluating, expanding, curating, and acting with MINT.** (a) How we build and expand MINT by first consulting a trained neural planning policy as an oracle, and then utilizing the LLM to curate the queries based on MINT and elicit human responses via natural-language interactions. (b) How MINT acts in the environment. AI agent implements the identified queries in its interaction with human in joint planning. The human responses are processed to produce a reduced knowledge gap $u_K$ at last, leading to an optimal action $a$ by maximizing $Q_\varphi^*(s, a)$ for all descriptors $\varphi \in \Phi_{u_K}$.

The estimated variance of Q-value is also used to quantify the uncertainty of the knowledge gap on planning outcomes during deployment. If the estimated variance $\tilde{\sigma}(s, a^*)$ is larger than a predefined threshold $\delta$ and a knowledge gap exists, e.g., an uncertain object detected by the perception module in planning, the agent will label it as *unresolved*, and switch to the MINT reasoning process. Otherwise, the unknown object is deemed to have little impact on the action selection or planning outcome, and the agent continues planning with the regular RL policy.

**Adapted DQN Training Paradigm** As mentioned above, we use a bootstrapped DQN architecture and training paradigm to estimate both $\mu_u(s, a) = \mathbb{E}_{\varphi \sim \Phi_u}[Q_\varphi^*(s, a)]$ and $\sigma_u^2(s, a)$ as the mean and variance of Q. For a single knowledge descriptor $\varphi \sim \Phi_u$, it is treated as an extra input appended to the state to generate Q-values. Besides, the descriptor space $\Phi_u$ itself can also be encoded as an input, yielding both the mean and the variance estimation for the Q-value on each action over the distribution of $\varphi$. During training, the ground-truth knowledge descriptor $\varphi^*$ is set to be fully observable. However, when collecting experience and adding an interaction sample to the replay buffer, an additional sample with a randomly generated knowledge gap $u$ will be added as well. This random knowledge gap will mask out part of the information on the original $\varphi$ and thus create a space $\Phi_u$ in the sample state, while keeping other interaction data such as the next state and rewards the same. Therefore, the trained model can be accurate under known descriptor $\varphi$ and also estimate the mean and variance of Q-values given a descriptor space $\Phi_u$ due to the knowledge gap $u$.

Next, we explain a bit more on how the variance is estimated. Initially, the uncertainty in the agent's policy can be further categorized into aleatoric uncertainty and epistemic uncertainty. The first case refers to the inherent randomness in transitions or rewards, while the latter arises from the agent's limited exploration of the environment. In this work, we focus on aleatoric uncertainty due to the knowledge gap, since in the view of the AI agent, the same knowledge gap $u$ can result in a variety of MDPs with different transition $T_\varphi$ and rewards $R_\varphi$ with $\varphi \sim \Phi_u$. Therefore, the only way to mitigate it is by obtaining more latent environmental knowledge from human inputs.

To decouple the two types of uncertainty and estimate aleatoric uncertainty only, we adopt an uncertainty-aware bootstrapped DQN architecture known as UA-DQN (Clements et al., 2019). As a method in distributional RL, the neural network is modified on the final layer to estimate a probability distribution of returns, parameterized by $N$ quantiles with value $q_i(s, a), i \in [1, N]$ as outputs. Thus, the aleatoric uncertainty can be estimated as $\tilde{\sigma}_{alea}^2(s, a) = cov_{i \sim \mathcal{U}\{1, N\}}(q_i(s, a|\theta_A), q_i(s, a|\theta_B))$, in which $\mathcal{U}$ is the uniform distribution while $\theta_A$ and $\theta_B$ are two samples of neural network parameter $\theta$ using MAP sampling (Pearce et al., 2018). The details of theories and implementation of UA-DQN are covered in the original paper, which is not the emphasis of our work and is thus omitted.

## 4.2 REASONING AND CURATING QUERIES WITH MINT

When the AI agent encounters an unseen or low-confidence object or an ambiguous task objective – whose unknown properties lead to substantial uncertainty in either the state transition $T$ or the reward function $R$, it triggers a reasoning process using MINT. It encompasses multiple steps including node representation, expansion, evaluation of knowledge gaps, and LLM-based curation and processing.

**Node Representation.**  Each node represents a knowledge gap $u$, with $\Phi_u$ explicitly represented as $\langle type, subtype, w_{min}, w_{max}\rangle$, summarizing the agent's current knowledge about the unknown object. Here, $type$ indicates whether the perturbance is on the state transition $T$ or the reward function $R$, while $subtype$ denotes the critical attribute of the values (*e.g.*, positive or negative rewards, deterministic or stochastic transitions). $w_{min}$ and $w_{max}$ are the minimum and maximum of the possible value, which is either the possibility parameter in $T_\varphi$ or the reward parameter in $R_\varphi$ for $\varphi \in \Phi_u$. For instance, in a deterministic and discrete environment, we normalize the reward range to $[0, 1]$ and bound the perturbation into the form of a set of half-blocked states $\bar{S}$. Thus, for any $\bar{s} \in \bar{S}$, the agent gets an immediate reward $r(\bar{s}) \in [0, 1]$ when the environment changes to $\bar{s}$, and in this case $w_{min}$ and $w_{max}$ provide the lower and upper bounds of $r(\bar{s})$ based on the agent's current knowledge about the object. Similarly, for the transition case, any state transition to $\bar{s} \in \bar{S}$ now has probability $p(\bar{s})$ to fail and remain in the original state, in which $w_{min}$ and $w_{max}$ will be the upper and lower bounds of $p(\bar{s})$ instead. Initially, the root node will be the maximal knowledge gap $u$ with $\Phi_u = \langle any, any, 0, 1\rangle$.

**Evaluation and Expansion.**  From the root node, the neural-symbolic information tree starts expanding and creating new branches in the breadth-first order. When visiting each node $u$, we use a modified neural network from the original deep Q-learning architecture that takes both $(s, a)$ and four elements in $\Phi_u$ as inputs, to estimate both the mean $\mu_u(s, a)$ and variance $\sigma_u^2(s, a)$ of $Q_\varphi^*(s, a), \varphi \in \Phi_u$ for current $s$ and $\forall a \in \mathcal{A}$. Thus, we can obtain the optimal action of this node as: $a_u^* = \arg\max_a \mu_u(s, a)$. Next, we compare the estimated standard deviation $\sigma_u(s, a_u^*)$ with the average return gap between the best action and other actions: $g(u) = \mu_u(s, a_u^*) - \max_{a \neq a_u^*} \mu_u(s, a)$. With $\lambda_g$ defined as a hyperparameter, if $g(u) \leq \lambda_g \sigma_u(s, a_u^*)$ and node $u$ has not reached the depth limit $T_D$, this node will be further expanded with new branches; otherwise, it will remain a leaf node. To expand a node $u$, the tree will create several new child nodes with hypothetical knowledge gap $u'$ based on the parent knowledge $u$, in the order of type, subtype and then value. If the type is unknown, then every new child node will set the knowledge with different possible types and leave other dimensions unchanged, similar to the subtype. If both type and subtype are filled, then there will be two new nodes evenly dividing the value range.

**LLM-Based Curation and Processing.**  After MINT is built, we encode it into a concise natural-language prompt and provide it to the LLM, which then (i) merges equivalent sub-trees, (ii) formulates an informative yes/no query for the human in each round, and (iii) updates the tree with the returned answers, and (iv) recursively repeats the process until the optimal action is identified in the tree or the queries reach maximum complexity.

Two types of merging operations are performed. First, if every leaf beneath a parent node leads to the same optimal action, then all leaves are folded into their parent, eliminating redundant queries. Second, distinct branches that ultimately associate with an identical action are integrated into a single disjunctive node with a joint condition. For instance, branches defined by $(type = 1, subtype = 1)$ and $(type = 2, subtype = 2)$ both having the optimal action $1$, are reorganized as a node under the root with the joint condition $(type = 1 \wedge subtype = 1) \vee (type = 2 \wedge subtype = 2)$. After pruning, the LLM is required to synthesize a binary question whose answer will maximize information gain on the optimal action candidates. This question will then be sent to humans and wait for the human answer. In our experiments, we use another LLM with full knowledge to automatically generate the yes/no answers. This response is then appended to the dialogue context and used to prune all branches inconsistent with that answer in both the original and the reorganized tree. If ambiguity in action selection remains, this procedure restarts with the updated tree and a new root.

## 4.3 THEORETICAL ANALYSIS

We show that MINT provides a guaranteed planning performance, by iteratively narrowing down the knowledge gap through interactions. To this end, we provide an upper bound on the return gap between MINT-based planning and an ideal case without the knowledge gap. We derive this bound by first defining the metric between different MDPs, then demonstrating the one-step Bellman bound on q-functions, inductively generalizing it to the Lipschitz continuity of optimal Q-functions, and finally proving the upper bound on returns for intermediate MDPs.

**Definition 4.1.** Consider two MDPs that differ slightly in their transition kernels and reward functions. Let the two MDPs be denoted as $M = (S, A, T, R, \gamma)$ and $\bar{M} = (S, A, T', R', \gamma)$, where $\gamma \in [0, 1)$. The pseudo-metric $\Delta_{s,a}(M, \bar{M})$ between these two processes at $(s, a) \in \mathcal{S} \times \mathcal{A}$ is defined as:

$$\Delta_{s,a}(M\|\bar{M}) = \min\{d_{s,a}(M\|\bar{M}), d_{s,a}(\bar{M}\|M)\} \tag{3}$$

Here, $d_{s,a}(\bar{M}\|M)$ is the MDP dissimilarity defined as the unique solution to the fixed-point equation for $d_{s,a}$:

$$d_{s,a} = |R(s,a) - R'(s,a)| + \gamma \sum_{s' \in \mathcal{S}} V^*(s')|T(s'|s,a) - T'(s'|s,a)| + \gamma \sum_{s' \in \mathcal{S}} T(s'|s,a) \max_{a'} d_{s',a'} \tag{4}$$

This definition is for discrete state spaces and can be straightforwardly extended to continuous state spaces. $\Delta_{s,a}$ is called a pseudo-metric since it doesn't obey the positive definiteness, *i.e.*, we have $\Delta_{s,a}(x, x) = 0, \forall x$ but $\Delta_{s,a}(x, y) = 0 \nRightarrow x = y$. Hence we use the term pseudo-Lipschitz continuity to denote that it's in the same form as Lipschitz continuity but built on a pseudo-metric. Before proving the continuity, we first provide a lemma on one-step Bellman bound.

**Lemma 4.2** (One-step Bellman bound). *With $\Gamma$ defined as the Bellman Operator on any function $Q : \mathcal{S} \times \mathcal{A} \to \mathbb{R}$ as:*

$$\Gamma Q(s,a) = R(s,a) + \gamma \sum_{s' \in \mathcal{S}} T(s'|s,a) \max_{a' \in \mathcal{A}} Q(s',a'), \tag{5}$$

*for any two MDPs $M$ and $\bar{M}$, if function $Q$ is already bounded by $\Delta_{s,a}(M, \bar{M})$, i.e., $|Q_M(s,a) - Q_{\bar{M}}(s,a)| \le \Delta_{s,a}(M, \bar{M})$, then we can guarantee:*

$$|\Gamma Q_M(s,a) - \Gamma Q_{\bar{M}}(s,a)| \le \Delta_{s,a}(M, \bar{M}). \tag{6}$$

Lemma A.2 is an inductive condition to prove the pseudo-Lipschitz continuity of the optimal Q-function $Q^*$, based on the fact that the optimal Q-function $Q^*$ is the unique fixed point of the Bellman equation and thus obtained by value iteration in Q-learning algorithms from identical initial values over different MDPs. Thus, by using induction, we can derive Lemma A.3 from Lemma A.2:

**Lemma 4.3** (Local pseudo-Lipschitz continuity of optimal Q-value). *For two MDPs $M, \bar{M}$, for all $(s, a) \in \mathcal{S} \times \mathcal{A}$, we have: $|Q_M^*(s,a) - Q_{\bar{M}}^*(s,a)| \le \Delta_{s,a}(M, \bar{M})$.*

Recall the extended MDP family defined previously. With this theorem we can provide an upper bound on the optimal Q-functions of any intermediate MDP between any two known MDPs $M_{\varphi_1}, M_{\varphi_2}$. Furthermore, by limiting the ground-truth descriptor $\varphi^*$ associated with the unknown knowledge gap $u$ between two descriptor samples $\varphi_1, \varphi_2$ with known estimated returns, we can prove an upper bound for the return gap under unknown $\varphi^*$ and its optimal policy $\pi_{\varphi^*}^*$.

**Theorem 4.4** (Upper bound of return for an unknown knowledge gap). *Given two MDPs $M_{\varphi_1}, M_{\varphi_2}$ with $\varphi_1, \varphi_2 \in \Phi$, for all $(s, a) \in \mathcal{S} \times \mathcal{A}$ and an unknown intermediate MDP $M_{\varphi^*}, \varphi^* = \lambda\varphi_1 + (1 - \lambda)\varphi_2, \lambda \in (0, 1)$, the upper bound $U$ on $J(\pi_{\varphi^*}^*|\varphi^*)$ can be defined as:*

$$J(\pi_{\varphi^*}^*|\varphi^*) \le U_{\varphi^*}(\varphi_1, \varphi_2) = \min\{J(\pi_{\varphi_1}^*|\varphi_1), J(\pi_{\varphi_2}^*|\varphi_2)\} + \Delta_{s_0,a_0}(M_{\varphi_1}, M_{\varphi_2}) \tag{7}$$

The theorem shows that by iteratively dividing and narrowing down the knowledge gap $u$ with MINT, we can in the end achieve a guaranteed return gap toward the ideal case. Additional interactions to further narrow the knowledge gap reduce the return gap $\Delta_{s_0,a_0}(M_{\varphi_1}, M_{\varphi_2})$ in the upper bound.

## 5 EXPERIMENTS

In this section, we present the environmental setting, baselines, evaluation metrics, and our numerical results in three environments with increasing realism. We show that MINT can efficiently elicit human inputs and leverage them to directly improve planning performance.

| Method | Version | No uncertainty | | 1 uncertain object | | 3 uncertain objects | | 5 uncertain objects | |
|---|---|---|---|---|---|---|---|---|---|
| | | Success% | Avg. Reward | Success% | Avg. Reward | Success% | Avg. Reward | Success% | Avg. Reward |
| LLM | GPT-4o | 100 | $9.26 \pm 0.42$ | 99 | $9.30 \pm 1.36$ | 96 | $8.75 \pm 2.72$ | 95 | $8.61 \pm 2.92$ |
| | o3-mini | 100 | $9.31 \pm 0.39$ | 99 | $9.29 \pm 1.40$ | 100 | $9.33 \pm 0.47$ | 99 | $8.09 \pm 3.83$ |
| | o3 | 100 | $9.29 \pm 0.44$ | 100 | $9.35 \pm 0.41$ | 100 | $9.35 \pm 0.48$ | 98 | $7.81 \pm 4.17$ |
| Pure RL | DQN | $100.0 \pm 0.0$ | $9.32 \pm 0.41$ | $83.6 \pm 1.8$ | $6.99 \pm 4.99$ | $65.0 \pm 7.9$ | $4.63 \pm 6.34$ | $15.2 \pm 6.2$ | $-2.01 \pm 4.52$ |
| | PPO | $100.0 \pm 0.0$ | $9.30 \pm 0.48$ | $98.2 \pm 1.0$ | $9.04 \pm 1.21$ | $89.8 \pm 3.1$ | $7.88 \pm 4.19$ | $69.2 \pm 13.0$ | $5.42 \pm 6.02$ |
| MINT | Limited | $100.0 \pm 0.0$ | $9.45 \pm 0.40$ | $99.6 \pm 0.5$ | $9.29 \pm 1.47$ | $100 \pm 0.0$ | $9.90 \pm 1.09$ | $97.0 \pm 0.9$ | $9.56 \pm 2.31$ |
| | Standard | $100.0 \pm 0.0$ | $9.47 \pm 0.33$ | $100.0 \pm 0.0$ | $9.75 \pm 0.66$ | $99.4 \pm 0.5$ | $9.71 \pm 1.88$ | $98.2 \pm 1.3$ | $9.69 \pm 1.91$ |

Table 1: The evaluation of different planning strategy in the MiniGrid environment. Here, the standard MINT does not have the limitation on the number of queries, while the limited version has a maximum number of queries for each unknown object. The results show that MINT significantly improve the planning performance with higher reward (primary planning objective) and improved success rate (narrowly defined as task completion), as compared to pure-LLM and pure-RL methods.

**Environmental Setting** We evaluate our method in three environments to demonstrate the performance, query cost, and generality. The first two are based on the publicly available environment MiniGrid (Chevalier-Boisvert et al., 2023) and Atari Pacman (Bellemare et al., 2013), in which we introduce the uncertainty by adding unknown objects/elements affecting observation, state transition, and reward functions. This object is represented as an encoded block in MiniGrid or a pure-color rectangular area in Atari Pacman, which can be a randomized reward/penalty, an obstacle that can be passed through, or a terminal with a random possibility to happen. In MiniGrid, the agent needs to navigate to a final goal with a high reward, while taking each step has a minor negative penalty as cost. In Atari Pacman, the reward function remains the same except for the uncertain area. For the final environment, we create a high-fidelity emergency response scenario based on the Nvidia Isaac platform, in which the drone agent is asked to rescue an injured person in an unseen warehouse environment via visual 3D reconstruction(using Gaussian splatting Chen & Wang (2025)) and planning. The AI agent encounters objects with low confidence (e.g., equipment on fire and boxes obfuscated by smoke) and must interact with humans to bridge the knowledge gap in planning, e.g., by asking whether to avoid a smoky area (i.e., uncertainty affecting transition probabilities) or to explore potential rescue targets in a separate room (i.e., uncertainty affecting reward functions).

| Method | Version | Return | Query Time |
|---|---|---|---|
| Pure RL | DQN | $271.3 \pm 21.7$ | - |
| | PPO | $325.0 \pm 34.5$ | - |
| Query-A | - | $422.1 \pm 24.7$ | $27.1 \pm 13.6$ |
| MINT | Limited | $411.1 \pm 28.8$ | $3.8 \pm 2.4$ |
| | Standard | $434.9 \pm 32.1$ | $6.8 \pm 5.5$ |

Table 2: Results on Atari Pacman. Query-A (with human-in-the-loop RL) asks for expert action when the variance is high.

| Method | Version | Target 1 | Target 2 |
|---|---|---|---|
| LLM | GPT-4o | 47 | 5 |
| | o3-mini | 89 | 11 |
| | o3 | 93 | 9 |
| MINT* | - | 100 | 95 |

Table 3: Success rates on NVIDIA Isaac. Target 1 and target 2 refers to rescuing the main target and a hidden target. MINT* uses LLM for both planning and interaction.

**Baselines and Metrics** In MiniGrid, we compare MINT with both pure-LLMs and pure-RL methods with human-in-the-loop to demonstrate that MINT combines the advantages of both sides – i.e., language-based interaction and reward optimization – for significantly better performance. These methods have no query mechanism and no knowledge about the uncertainty. The pure-LLM method gets a prompt with encoded environment observation, and plans the entire path at once. Here, the metrics are the success rate and the average rewards for each episode. An episode is considered successful if the agent reaches the goal area within limited steps. The observation in Atari Pacman is raw pixels and hard to encode as natural language; therefore, we only compare the pure-RL and

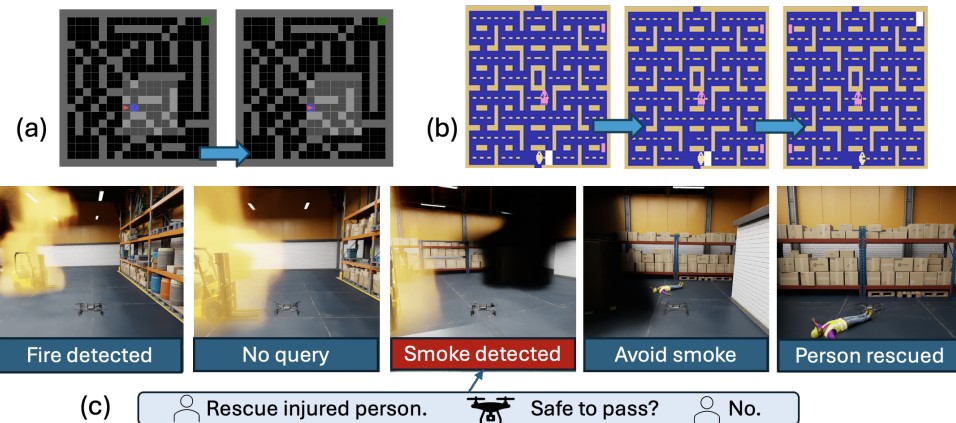

Figure 2: Illustrations of how MINT acts in all 3 environments. (a) The agent faces unknown objects in MiniGrid and curates queries about its impact on transition; (b) The agent in Atari Pacman faces unseen targets (white) and curates queries about its impact on rewards; and (c) The agent in Isaac Search and Rescue reasons about the smoke, interacts with human, and plans its path accordingly.

Query-baseline with MINT and use average reward and query times as metrics. The Query-baseline is the classical query method that asks for optimal action when variance is high. In Isaac environments, we remove the RL part of MINT and only use a fixed tree structure and the LLM to plan the path directly; thus, we show that MINT can also be generalized to non-RL planning frameworks in realistic scenarios. We calculate the success rate for rescuing all targets without crashing into objects and use it as the metric. For each environment, we evaluate for 100 runs and calculate the average and standard deviation. For RL methods and MINT involving training, we trained for 5 random seeds.

**Results Analysis** We first report the performance results of the MiniGrid environment with average and standard deviation in Table 1. Both LLM and RL result in low rewards as our main evaluation metric for task performance. In particular, recent LLM methods most likely generate code using classical navigation algorithms such as A* while ignoring the knowledge gap and uncertainty, and then execute it to get results. Although the success rate (narrowly defined as task completion) is high, such a method is considered a conservative strategy since the hidden bonus from the uncertainty is not utilized. Besides, pure RL methods perform poorly since their neural network can be unpredictable with uncertain objects. In comparison, MINT iteratively narrows down the knowledge gap through queries, thus obtaining significantly improved performance. Also, even with a very limited number of queries, the performance is still competitive, since the symbolic tree in MINT is used to reason and optimize query strategy. Besides, results in Atari Pacman shown in Table 2 demonstrate that MINT can also be integrated into environments with complicated observation spaces, and it reaches competitive performance compared with traditional human-in-the-loop RL (HITL-RL) methods requiring expert actions, while significantly dropping the need for queries. Finally, with the results in the Isaac environment presented in Table 3, we show that MINT can be generalized to realistic planning problems with continuous control commands. While LLM-based planning strategies often have difficulty dealing with uncertain threats and hidden targets in this search and rescue task, MINT solves this problem by reasoning and curating queries, thus largely outperforming the baselines.

## 6 CONCLUSION

MINT introduces a novel approach to optimize human elicitation in language-based planning tasks. By consulting a neural planning policy and reasoning the impact of knowledge gaps, it systematically narrows down the knowledge gap through iterative query curation and thus resolves the decision-making uncertainty. MINT is shown to provide guarantees on planning performance through a local Lipschitz continuity property. Empirically, we demonstrated that MINT substantially outperforms existing pure-LLM and pure-RL baselines with human-in-the-loop, across diverse benchmarks including MiniGrid, Atari Pacman, and the realistic Isaac scenario. MINT leads to novel solutions for human-AI collaborative planning through neuro-symbolic reasoning and self-play.

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

## A  APPENDIX

### A.1  USAGE OF LARGE LANGUAGE MODELS

The Large Language Models are used as a significant part of the methodology proposed in this paper. Nevertheless, they are not used for research ideation, derivations, proofs, experimental design, data analysis, or writing to the extent that they could be regarded as a contributor to authorship or a significant contribution under the conference policy.

### A.2  THEORETICAL DERIVATIONS AND PROOFS

In this section, we provide detailed theoretical derivations and proofs of Lemma 4.2, Lemma 4.3, which are mainly inspired by previous works (Lecarpentier et al., 2021), and thus leading to our main theorem 4.4. Before that, we need to first prove that Definition 4.1 is valid, *i.e.*, there is a unique solution for $d_{s,a}$.

**Lemma A.1.** *Given two MDPs $M = (\mathcal{S}, \mathcal{A}, T, R, \gamma)$ and $\bar{M} = (\mathcal{S}, \mathcal{A}, T', R', \gamma)$ that differ slightly in their transition kernels and reward functions, the following equation on $d : \mathcal{S} \times \mathcal{A} \to \mathbb{R}$ is a fixed-point equation admitting a unique solution for any $(s, a)$:*

$$d_{s,a} = |R(s,a) - R'(s,a)| + \gamma \sum_{s' \in \mathcal{S}} V_{\bar{M}}^*(s')|T(s'|s,a) - T'(s'|s,a)| + \gamma \sum_{s' \in \mathcal{S}} T(s'|s,a) \max_{a' \in \mathcal{A}} d_{s',a'}$$

*Proof.* We denote

$$D_{s,a}(M\|\bar{M}) = |R(s,a) - R'(s,a)| + \gamma \sum_{s' \in \mathcal{S}} V_{\bar{M}}^*(s')|T(s'|s,a) - T'(s'|s,a)|.$$

Let $\mathcal{L}$ be the functional operator that maps any function $d \in \mathcal{F}(\mathcal{S} \times \mathcal{A}, \mathbb{R})$ as follows:

$$\mathcal{L}d_{s,a} = D_{s,a}(M\|\bar{M}) + \gamma \sum_{s' \in \mathcal{S}} T(s'|s,a) \max_{a' \in \mathcal{A}} d_{s',a'}.$$

Therefore, for any two functions $f$ and $g$ in $\mathcal{F}(\mathcal{S} \times \mathcal{A}, \mathbb{R})$ and any $(s, a) \in \mathcal{S} \times \mathcal{A}$, we have:

$$\mathcal{L}f_{s,a} - \mathcal{L}g_{s,a} = \gamma \sum_{s' \in \mathcal{S}} T(s'|s,a) \left( \max_{a' \in \mathcal{A}} f_{s',a'} - \max_{a' \in \mathcal{A}} g_{s',a'} \right)$$

$$\leq \gamma \sum_{s' \in \mathcal{S}} T(s'|s,a) \max_{a' \in \mathcal{A}}(f_{s',a'} - g_{s',a'})$$

$$\leq \gamma \|f - g\|_\infty.$$

Hence, we have $\|\mathcal{L}f - \mathcal{L}g\|_\infty \leq \gamma \|f - g\|_\infty$. Since $\gamma < 1$, $\mathcal{L}$ is a contraction mapping in the complete and non-empty metric space $(\mathcal{F}(\mathcal{S} \times \mathcal{A}, \mathbb{R}), \|\cdot\|_\infty)$. By directly applying the Banach fixed-point theorem, we can conclude that the previous equation on $d_{s,a}$ admits a unique solution. $\square$

Next, we move on to prove the one-step Bellman bound in the main text.

**Lemma A.2** (One-step Bellman Bound). *With $\Gamma$ defined as the Bellman Operator on any function $Q : \mathcal{S} \times \mathcal{A} \to \mathbb{R}$ as:*

$$\Gamma Q(s,a) = R(s,a) + \gamma \sum_{s' \in \mathcal{S}} T(s'|s,a) \max_{a' \in \mathcal{A}} Q(s',a'), \tag{8}$$

*for any two MDPs $M$ and $\bar{M}$, if function $Q$ is already bounded by $\Delta_{s,a}(M, \bar{M})$, i.e., $|Q_M(s,a) - Q_{\bar{M}}(s,a)| \leq \Delta_{s,a}(M, \bar{M})$, then we can guarantee:*

$$|\Gamma Q_M(s,a) - \Gamma Q_{\bar{M}}(s,a)| \leq \Delta_{s,a}(M, \bar{M}). \tag{9}$$

*Proof.* Since $\Delta_{s,a}(M, \bar{M}) = \min\{d_{s,a}(M\|\bar{M}), d_{s,a}(\bar{M}\|M)\}$, we can separately prove that $|\Gamma Q_M(s,a) - \Gamma Q_{\bar{M}}(s,a)|$ is less than or equal to both $d_{s,a}(M\|\bar{M})$ and $d_{s,a}(\bar{M}\|M)$, and then

summarize them to get the conclusion.

$$|\Gamma Q_M(s,a) - \Gamma Q_{\bar{M}}(s,a)|$$

$$= \left| R(s,a) - R'(s,a) + \gamma \sum_{s' \in \mathcal{S}} \left[ T(s'|s,a) \max_{a' \in \mathcal{A}} Q_M(s',a') - T'(s'|s,a) \max_{a' \in \mathcal{A}} Q_{\bar{M}}(s',a') \right] \right|$$

$$\leq |R(s,a) - R'(s,a)| + \gamma \sum_{s' \in \mathcal{S}} \left| T(s'|s,a) \max_{a' \in \mathcal{A}} Q_M(s',a') - T'(s'|s,a) \max_{a' \in \mathcal{A}} Q_{\bar{M}}(s',a') \right|$$

$$\leq |R(s,a) - R'(s,a)| + \gamma \sum_{s' \in \mathcal{S}} \max_{a' \in \mathcal{A}} Q_{\bar{M}}(s',a') |T(s'|s,a) - T'(s'|s,a)|$$

$$+ \gamma \sum_{s' \in \mathcal{S}} T(s'|s,a) \left| \max_{a' \in \mathcal{A}} Q_M(s',a') - \max_{a' \in \mathcal{A}} Q_{\bar{M}}(s',a') \right|$$

$$\leq |R(s,a) - R'(s,a)| + \gamma \sum_{s' \in \mathcal{S}} V_{\bar{M}}^*(s') |T(s'|s,a) - T'(s'|s,a)|$$

$$+ \gamma \sum_{s' \in \mathcal{S}} T(s'|s,a) \max_{a' \in \mathcal{A}} |Q_M(s',a') - Q_{\bar{M}}(s',a')|$$

$$\leq D_{s,a}(M\|\bar{M}) + \gamma \sum_{s' \in \mathcal{S}} T(s'|s,a) \max_{a' \in \mathcal{A}} \Delta_{s',a'}(M,\bar{M})$$

$$\leq D_{s,a}(M\|\bar{M}) + \gamma \sum_{s' \in \mathcal{S}} T(s'|s,a) \max_{a' \in \mathcal{A}} d_{s',a'}(M\|\bar{M})$$

$$\leq d_{s,a}(M\|\bar{M}). \qquad \dots \qquad \text{Using Lemma A.1}$$

$$\square$$

Similarly, by exchanging the order of $\Gamma Q_M(s,a)$ and $\Gamma Q_{\bar{M}}(s,a)$, we can also derive:

$$|\Gamma Q_M(s,a) - \Gamma Q_{\bar{M}}(s,a)| \leq d_{s,a}(\bar{M}\|M)$$

By combining these two inequations, we can finally prove that:

$$|\Gamma Q_M(s,a) - \Gamma Q_{\bar{M}}(s,a)| \leq \Delta_{s,a}(M,\bar{M}).$$

With Lemma A.2, the local pseudo-Lipschitz continuity of the optimal Q-function can therefore be derived from induction.

**Lemma A.3** (Local pseudo-Lipschitz continuity of Optimal Q-value). *For two MDPs $M, \bar{M}$, for all $(s,a) \in \mathcal{S} \times \mathcal{A}$, we have: $|Q_M^*(s,a) - Q_{\bar{M}}^*(s,a)| \leq \Delta_{s,a}(M,\bar{M})$.*

*Proof.* Suppose we are using the same value iteration algorithm for both $M$ and $\bar{M}$. Then, the initial Q-function of each state and action should be exactly the same:

$$|Q_M^0(s,a) - Q_{\bar{M}}^0(s,a)| = 0 \leq \Delta_{s,a}(M,\bar{M}).$$

In each value iteration, the q-function is updated by the Bellman operator:

$$Q_M^{n+1}(s,a) = \Gamma Q_M^n(s,a) = R(s,a) + \gamma \sum_{s' \in \mathcal{S}} T(s'|s,a) \max_{a \in \mathcal{A}} Q_M^n(s',a'), \forall n \in \mathbb{N},$$

Hence, by applying induction and Lemma A.2, we have:

$$|Q_M^n(s,a) - Q_{\bar{M}}^n(s,a)| \leq \Delta_{s,a}(M,\bar{M}), \forall n \in \mathbb{N}.$$

The value iteration converges to the optimal Q-function: $\lim_{n \to \infty} Q_M^n(s,a) = Q_M^*(s,a)$, therefore we can conclude that $Q^*$ is pseudo-Lipschitz continuous in the local MDP space with:

$$|Q_M^*(s,a) - Q_{\bar{M}}^*(s,a)| \leq \Delta_{s,a}(M,\bar{M}).$$

$$\square$$

With Lemma A.3, we can deliver the proof of the main theorem in this paper as the upper bound of return for an unknown knowledge gap.

**Theorem A.4** (Upper bound of return for an unknown knowledge gap). *Given two MDPs $M_{\varphi_1}, M_{\varphi_2}$ with $\varphi_1, \varphi_2 \in \Phi$, for all $s_0 \in \mathcal{S}$ and an unknown intermediate MDP $M_{\varphi^*}, \varphi^* = \lambda\varphi_1 + (1-\lambda)\varphi_2, \lambda \in (0,1)$, the upper bound $U$ on $J(\pi_{\varphi^*}^* | \varphi^*)$ can be defined as:*

$$J(\pi_{\varphi^*}^* | \varphi^*) \leq U_{\varphi^*}(\varphi_1, \varphi_2) = \min\{J(\pi_{\varphi_1}^* | \varphi_1), J(\pi_{\varphi_2}^* | \varphi_2)\} + \Delta_{s_0, a_0}(M_{\varphi_1}, M_{\varphi_2}), \quad (10)$$

*in which $a_0 = \max_{a \in \mathcal{A}} \Delta_{s_0, a}(M_{\varphi_1}, M_{\varphi_2})$*

*Proof.* Recalling the definition of optimal Q-function: $J(\pi_{\varphi^*}^* | \varphi^*) = \max_{a \in \mathcal{A}} Q_{M_{\varphi^*}}^*(s_0, a)$, we have:

$$
\begin{aligned}
|J(\pi_{\varphi^*}^* | \varphi^*) - J(\pi_{\varphi_1}^* | \varphi_1)| &= |\max_{a \in \mathcal{A}} Q_{M_{\varphi^*}}^*(s_0, a) - \max_{a \in \mathcal{A}} Q_{M_{\varphi_1}}^*(s_0, a)| \\
&\leq \max_{a \in \mathcal{A}} |Q_{M_{\varphi^*}}^*(s_0, a) - Q_{M_{\varphi_1}}^*(s_0, a)| \\
&\leq \max_{a \in \mathcal{A}} \Delta_{s_0, a}(M_{\varphi^*}, M_{\varphi_1}) \\
&\leq \max_{a \in \mathcal{A}} \Delta_{s_0, a}(M_{\varphi_1}, M_{\varphi_2})
\end{aligned}
$$

Similarly, we can also prove that $|J(\pi_{\varphi^*}^* | \varphi^*) - J(\pi_{\varphi_2}^* | \varphi_2)| \leq \max_{a \in \mathcal{A}} \Delta_{s_0, a}(M_{\varphi_1}, M_{\varphi_2})$. Combining these two, we can conclude the proof. □

### A.3 DETAILED ALGORITHMIC IMPLEMENTATION

#### A.3.1 PSEUDOCODE

The overview pseudocode of MINT is divided into the training phase and deployment phase, demonstrated in Algorithm 1 and Algorithm 2, respectively. The details of UA-DQN training can be found in its original paper (Clements et al., 2019), while the node expansion and curation details are mainly explained in the main text.

---
**Algorithm 1:** MINT Training Phase

---
Initialize replay buffer $B$, UA-DQN $Q_\theta$
**for** $i = 0 \rightarrow N_e$ **do**
  Reset the environment $M_\varphi$ with a randomized descriptor $\varphi$. Get initial state $s = s_0$ from $M_\varphi$
  **while** *done is False* **do**
    Choose action $a$ according to policy derived from $Q_\theta(s, a | \varphi)$
    Execute $a$, observe reward $r$, next state $s'$, terminal signal *done*
    Add $(s, \varphi, a, r, s')$ to B
    Randomly generate partial knowledge gap $u$ masking $\varphi \rightarrow \Phi_u$
    Generate partial gap u, create masked descriptor space $\Phi_u$
    Add additional sample $(s, \Phi_u, a, r, s')$ to $B$
  Update $\theta$ with $\mathcal{B}$ using the loss function defined in UA-DQN

---

#### A.3.2 HYPERPARAMETERS

Table 4 shows the major hyperparameters we use in MINT experimental settings.

#### A.3.3 LLM PROMPTS

Here we present LLM prompt examples for branch merging, query curation, and tree updating in Listing 1, Listing 2, and Listing 3, respectively. The response of each LLM will be pre-processed and then encoded for usage.

```
Given the following tree structure:

Node 0 (root): {Type=Unknown, Subtype=Unknown, Value range:[0,1], Optimal
    Action:0, Child: Node 1, Node 2}
```

---

**Algorithm 2:** MINT Deployment Phase

---

**while** *done is False* **do**
    Get current state $s$ from the environment.
    **if** *no uncertain object detected* **then**
        $a^* = \arg\max_a Q_\theta(s, a|\varnothing)$
        Execute $a^*$, continue
    **if** *new uncertain object detected* **then**
        Construct root node from initial gap $u_0$ and current state $s$
        $u = u_0$
    **while** *u exists* **do**
        Calculate $\mu_u(s, a), \sigma_u^2(s, a)$ for each $a$ using $Q_\theta$
        $a_u^* = \arg\max_a \mu_u(s, a)$
        $g(u) = \mu_u(s, a_u^*) - \max_{a \neq a_u^*} \mu_u(s, a)$
        **if** $g(u) \leq \lambda_g \sigma_u(s, a_u^*)$ *and not reaching depth limit* **then**
            Expand node $u$
        Move to next node $u$ using BFS.
    **for** $k = 0, 1, \cdots, K$ **do**
        sample $\varphi_1, \varphi_2$ as the boundary of $\Phi_{u_k}$
        **if** $\arg\max_a Q_\theta(s, a|\varphi_1) = \arg\max_a Q_\theta(s, a|\varphi_2)$ **then**
            Break
        $q_k$ = LLMCuration($u_k$)
        Get human answer $y_k$ from $q_k$
        Update $u_k$ to $u_{k+1}$ using $q_k$ and $y_k$ via $LLM$
    Execute $a^* = \arg\max_a \mu_{u_k}(s, a)$.

---

| Hyperparameters | Values |
|---|---|
| Learning Rate $l_r$ | 1e-4 |
| Epsilon $\varepsilon$ | 0.05 |
| Batchsize | 256 |
| Discount factor($\gamma$) | 0.99 |
| Number of quantiles | 50 |
| Replay Buffer Size | 1e6 |
| $\lambda_d$ | 1.0 |
| Depth Limit $T_D$ | 5 |
| Maximum Number of Queries $K$ | 5 |
| LLM Temperature | 0.5 |
| Episode Length(MiniGrid) | 400 |
| Training Episode | 5e7 |

Table 4: Hyperparameters in MINT

```
Node 1: {Type=Transition, Subtype=Unknown, Value range:[0,1], Optimal
    Action:1, Child:Node 3, Node 4, Node 5}
...

Identify redundant branches or leaves that leading to identical optimal
    action. Output the tree structure with the same format.
```

Listing 1: Branch Merging Prompt Example

```
Given the current symbolic tree structure:

Node 0 (root): {Type=Unknown, Subtype=Unknown, Value range:[0,1], Optimal
    Action:0, Child: Node 1, Node 2}
Node 1: {Type=Transition, Subtype=Unknown, Value range:[0,1], Optimal
    Action:1, Child:Node 3, Node 4, Node 5}
Node 2: {Type=Reward, Subtype=Unknown, Value range:[0,1], Optimal Action
    :0, Child:None}
Node 3:{Type=Transition, Subtype= determined, Value range:[1,1], Optimal
    Action:1, Child:None}
Node 4:{Type=Transition, Subtype= determined, Value range:[0,0], Optimal
    Action:0, Child:None}
Node 5:{Type=Transition, Subtype= stochastic, Value range:[0,1], Optimal
    Action:0, Child:None}

Formulate a single yes/no question based on "Type" to resolve the
    uncertainty in optimal action. Your question should have the maximal
    information gain and divide the original tree into two sub-trees with
     nearly consistent optimal actions.
```

Listing 2: Curation Prompt Example

```
The current symbolic tree structure is as follows:

Node 0 (root): {Type=Unknown, Subtype=Unknown, Value range:[0,1], Optimal
    Action:0, Child: Node 1, Node 2}
Node 1: {Type=Transition, Subtype=Unknown, Value range:[0,1], Optimal
    Action:1, Child:Node 3, Node 4, Node 5}
Node 2: {Type=Reward, Subtype=Unknown, Value range:[0,1], Optimal Action
    :0, Child:None}
Node 3:{Type=Transition, Subtype= determined, Value range:[1,1], Optimal
    Action:1, Child:None}
Node 4:{Type=Transition, Subtype= determined, Value range:[0,0], Optimal
    Action:0, Child:None}
Node 5:{Type=Transition, Subtype= stochastic, Value range:[0,1], Optimal
    Action:0, Child:None}

Given the fact that the answer to the question "Is the uncertainty about
    a Transition parameter?" is "yes", remove branches inconsistent with
    this response and clearly state the pruned tree structure.
```

Listing 3: Tree Updating Prompt Example

## A.4 ADDITIONAL EXPERIMENTAL DETAILS AND RESULTS

### A.4.1 ENVIRONMENT DETAILS

In this paper, we use three different environments:MiniGrid, Atari Pacman, and NVIDIA Isaac-based environments. The screenshots of each environment are presented in Figure 3.

MiniGrid is a maze-like environment, with the agent starting from the top left. The goal of the agent is to reach the goal, marked as the green block in the top right, within minimal steps. When the agent gets to the green block, it will receive a reward of 10. Otherwise, the agent will get a reward of -0.1 for each step. The episode ends when the agent reaches the goal or maximum steps are used. The

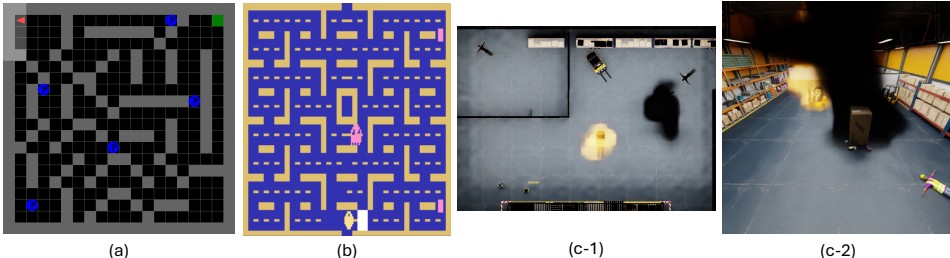

Figure 3: Screenshots of the environments used in this paper. (a)MiniGrid (b)Atari Pacman (c-1) an overview of NVIDIA Isaac environment (c-2) an example of drone view in Isaac environment.

agent can go to neighboring blocks in four directions, which is also the action space. Grey blocks refer to a wall that the agent cannot stand on or go through. There are 1-5 uncertain objects in certain fixed locations of the environment, marked as blue blocks, which either have an effect on transition or reward with a random value. The observation space of the agent will be its local view ($7 \times 7$ square in front of it) and current location.

The Atari Pacman environment is mainly based on its original game setting. However, we inject an uncertain object marked as the white rectangle in the raw frames. This uncertain object either has an effect on the transition or reward with a random value. By using template matching, we can locate the agent and thus manipulate the reward function and transition when there is overlap. Besides, we use the signal function to regularize the original reward in Atari environment into $\{0, 1\}$.

Finally, we create a high-fidelity emergency response scenario based on the Nvidia Isaac platform, in which a drone is asked to rescue an injured person in an unseen warehouse environment via visual 3D reconstruction, as shown in Figure 3 (c-1). Inside this warehouse, there is a fired forklift, an obstacle hidden by the smoke, and an extra injured person in another room. A VLM model will first scan the surroundings and execute 3D reconstruction, then locate the coordinates of each object. Next, the drone controlled by LLM will first plan a path represented in a sequence of coordinates to achieve the specific goals, then execute low-level control commands to move along this path.

### A.4.2 COMPUTATIONAL RESOURCES

All of the experiments in this paper are conducted on a server with an AMD EPYC 7513 128-Core Processor CPU and an NVIDIA RTX A6000 GPU.

### A.4.3 PERFORMANCE COMPARISON WITH INFERENTIAL LLMS

One of the drawbacks of the pure LLM method in MiniGrid environment is that LLM can be overly conservative in dealing with the uncertainties. It calls classical path-finding algorithms to reach the target and likely avoids the uncertainty blocks, even though it can be a positive reward or have little impact on transition, yielding relatively low reward compared to MINT, despite a good success rate. On the other hand, Query-A asks for optimal action whenever the variance is high. It assumes that a human user always gives the optimal action recommendation, thus achieving a similar reward as MINT. But human response is limited to the current state/choices in Query-A, unlike MINT, which can learn new knowledge with each response and apply it to reason new states/choices.

The point of comparing pure LLM approaches with MINT is that MINT combines neural policy (UA-DQN) with LLM to provide the optimal action based on variance estimates and human binary queries. This hybrid ensures robustness where pure LLMs fail.

Currently, we're using GPT-4o inside MINT in MiniGrid and Atari environments (will clarify this in the revision), and comparing it with pure LLM, also using GPT-4o and other advanced versions. Since MINT introduces a neuro-symbolic approach, we feel this comparison can demonstrate the advantage of MINT's neuro-symbolic approach. In the comparison, LLM methods require a full view of the environment (except details about the uncertainty) to plan the path. Compared with advanced models (o3 and o3-mini), MINT achieves comparative results by only using a low-level LLM (4o)

and partial observation. We also conducted experiments using a stronger LLM inside MINT, i.e., o3 with long-form and multi-step reasoning capability, with results in Table 5.

| LLMs | MiniGrid | | | Atari |
|---|---|---|---|---|
| | 1 object | 3 object | 5 object | |
| 4o Success% | $99.6 \pm 0.5$ | $100 \pm 0$ | $97.0 \pm 0.9$ | - |
| 4o Avg. Reward | $9.29 \pm 1.47$ | $9.90 \pm 1.09$ | $9.56 \pm 2.31$ | $411.1 \pm 28.8$ |
| o3 Success% | $100 \pm 0$ | $100 \pm 0$ | $98.6 \pm 1.2$ | - |
| o3 Avg. Reward | $9.91 \pm 0.48$ | $9.88 \pm 0.91$ | $9.76 \pm 1.37$ | $428.7 \pm 30.6$ |

Table 5: Comparative results with different models used in MINT.

The result shows that using an advanced LLM doesn't bring much improvement over the original MINT, since the performance of MINT with 4o is already close to the optimal. However, since o3 is a long, multi-step reasoning model with an explicit chain of thoughts, the time cost of using such an advanced LLM is huge, demonstrating another advantage of MINT in saving the time cost.

### A.4.4 UNCERTAINTY DECLINE ANALYSIS

We show the results of uncertainty decreasing over queries with human interactions in Table 6. The experiment is taken in the MiniGrid Environment, measuring the average uncertainty ( estimated by the variance of Q-value) after a different number of queries.

| Avg. Uncertainty / Queries | 0 | 1 | 2 | 3 | 4 |
|---|---|---|---|---|---|
| 5 objects | 6.50 | 5.36 | 4.13 | 4.07 | 3.88 |
| 3 objects | 5.73 | 4.14 | 3.79 | 3.13 | 2.93 |
| 1 object | 1.37 | 0.40 | 0.05 | 0.06 | 0.04 |

Table 6: Uncertainty Decline Analysis in the MiniGrid Environment.

From the results, the uncertainty decreases most at the first two queries, which meets our expectation that MINT will generate the query that maximizes the information gain. Besides, in the 1-object scene, the average uncertainty drops to nearly zero after only two queries, while with increasing uncertainties, more queries are required.

