# OpenReview forum: "MINT: Minimal Information Neuro-Symbolic Tree for Objective-Driven Knowledge-Gap Reasoning and Active Human Elicitation"
_ICLR.cc/2026/Conference — Submitted to ICLR 2026_

### Official Review · Reviewer_fY71 · 2025-10-24

**Soundness:** 3
**Presentation:** 3
**Contribution:** 3
**Rating:** 6
**Confidence:** 3

**Summary:**

The paper studies human-AI joint planning when the agent lacks key task facts. It formalizes a “knowledge gap” as a neat set of feasible task descriptors that induce a family of MDPs. The method, MINT, builds a small tree over these gaps, scores each node by estimated impact on the optimal action, and periodically asks the human a single yes/no question that prunes the tree. Questions are curated by an LLM to be concise and action-relevant. There is a local continuity bound that links smaller gaps to smaller value differences. Experiments on MiniGrid, a modified Atari setting, and an Isaac-Gym rescue scenario show strong returns with very few queries.

**Strengths:**

- Clear and elegant problem framing for planning-time human queries rather than training-time feedback. Binary questions keep human effort minimal while still moving the plan.
- Concrete gap representation and a simple tree procedure that is easy to follow and implement.
- Sensible query trigger + theoretical support: only ask when the current uncertainty can change the action and a pseudo-metric on MDPs and a continuity style bound that justifies stopping early.
- Empirics show near-expert performance with tiny query budgets on diverse tasks. And ablations around tree growth and stop rules are interpretable for practitioners.

**Weaknesses:**

- The gap parameterization seems hand-crafted. It is unclear how to scale the type-subtype-value scheme when unknowns are high dimensional or compositional.
- The uncertainty signal relies on Q-value variance from a bootstrapped policy. There is limited analysis of calibration, failure modes, or alternatives.
- LLM involvement in merging and question wording may add prompt sensitivity and non-determinism. Limited study of how this affects later plans. Human factors are not deeply evaluated. There is no user study on cognitive load, latency, or error rates when humans answer quickly or incorrectly (or in other words, what if human made mistakes).
- Detection of “uncertain objects” and the mapping from raw observations to gap descriptors is under-specified for real robotics or vision-heavy tasks.

**Questions:**

- Why tree structures are chosen for this work? Can we extend it to more generalizable representations like graphs?
- How robust is performance if the human sometimes answers incorrectly or says “I do not know”? Can MINT detect and recover from inconsistent answers?
- Can you replace the LLM merging and question curation with a rule-based variant and report deltas, so we know how much the LLM contributes? In other words, I personally think it would be better if the usage of LLMs is justified and tested.
- How do you construct gap descriptors from raw observations in more complex domains without oracle metadata? Can the descriptor be learned rather than hand-typed?
- What happens when there are multiple interacting unknowns that affect the same action choice? Does the tree branch order bias the questions and outcomes?

---

> ### Author Response · Authors · 2025-11-20
>
> > Weakness: The gap parameterization seems hand-crafted...
>
> Our theoretical framework only requires that the knowledge gap be represented as a bounded descriptor $\phi \in \Phi_u$ such that each $\phi$ defines a valid MDP $M_\phi$.
> The particular choice ⟨type, subtype, w⟩ in Sec. 4.2 is a minimal instantiation tailored to our environments. It allows us to explicitly control whether uncertainty lives in rewards vs transitions and whether it is “good/bad / terminal.” It keeps the tree shallow so we can highlight how query optimization, rather than sheer depth, drives efficiency.
>
> In the future work, we will:
> Emphasize that MINT is agnostic to the specific encoding of $\phi$, and research on learned descriptors, e.g., mapping object-level perceptual features into a low-dimensional latent via an encoder (like symbolic visual RL / neuro-symbolic RL) and using that latent as $\phi$, so that “hand-typing” is no longer necessary.
>
> > Weakness: The uncertainty signal relies on Q-value variance from a bootstrapped policy...
>
> We chose UA-DQN because it explicitly separates aleatoric uncertainty (due to unknown transitions/rewards) from epistemic uncertainty, and we use $\sigma_u^2(s,a)$ only as a heuristic to decide when to expand nodes or trigger MINT.
> Our theoretical bound (Theorem 4.4) depends only on the descriptor partition, not on the exact uncertainty estimator.
>
> > Weakness: Detection of “uncertain objects” and the mapping from raw observations ...
> > Questions: How do you construct gap descriptors from raw observations in more complex domains without oracle metadata? Can the descriptor be learned rather than hand-typed?
>
> As noted for Reviewer 6Xak, in the current benchmarks the mapping from raw observations to “uncertain objects” and gap descriptors is provided by the simulator:
>
> MiniGrid and Atari: we programmatically insert uncertain regions with known location and object ID; the raw observation is augmented with this metadata.
>
> Isaac: The simulator provides object IDs and detection confidences from the 3D reconstruction pipeline, and we treat low-confidence objects as candidates for knowledge gaps.
>
> The present paper focuses on reasoning and query selection given such candidate gaps, not on perception.
>
> > Questions: Why tree structures are chosen for this work? Can we extend it to more generalizable representations like graphs?
>
> It’s easy to structure binary questions: each edge corresponds to a yes/no query that partitions $\Phi_u$ into child regions. The tree induces a hierarchical partition of $\Phi_u$, which directly matches our theoretical analysis: Theorem 4.4 bounds the return gap between extreme descriptors in each final leaf. Trees are easy to prune and merge via LLMs (or rules), and the complexity of self-play over possible query sequences is manageable.
>
> In principle, MINT can be extended to DAG or graph structures where nodes represent overlapping constraints on $\phi$, as long as each region is associated with a bounded descriptor set; our pseudo-Lipschitz arguments still apply to each region.
>
> > Weakness: LLM involvement in merging and question wording...
>
> > Questions: Can you replace the LLM merging and question curation with a rule-based variant and report deltas, so we know how much the LLM contributes? In other words, I personally think it would be better if the usage of LLMs is justified and tested.
>
> Thanks for this suggestion. We'll add an additional ablation study on LLM merging and question curation in the revision.
>
> > Questions: How robust is performance if the human sometimes answers incorrectly or says “I do not know”? Can MINT detect and recover from inconsistent answers?
>
> Typically, MINT will not ask repeated questions. (Always trust the first answer, but it’s easy to extend to be able to detect and recover from inconsistent answers)
>
> Please check our response to Reviewer 6Xak as well.
>
> > What happens when there are multiple interacting unknowns that affect the same action choice? Does the tree branch order bias the questions and outcomes?
>
> In our current experiments, multiple uncertain objects are designed to be independent: each affects distinct regions or rewards, so their combined effect is additive.
>
> We handle this by running one MINT tree per object, resolving them sequentially. Thus, there is no interaction between trees in the current benchmarks, and the branch order within a tree only affects which aspect of that object we ask about first, not the final optimal action (since we continue splitting until $\sigma_u(s, a^*)$ is small or the depth limit is reached). To enable multiple interacting unknowns, we can establish a joint descriptor φ that encodes multiple objects together, and possibly a more general graph structure (as you suggested), to avoid strong order biases. We see this as an exciting direction for extending MINT to more complex, interacting uncertainties.

---

### Official Review · Reviewer_6Xak · 2025-10-30

**Soundness:** 2
**Presentation:** 2
**Contribution:** 3
**Rating:** 4
**Confidence:** 4

**Summary:**

The work focuses on human-AI collaboration, where incomplete information is presented in planning in the real world. To address this knowledge gap, the work first trains a DQN to quantify the uncertainty of selecting consistent actions under different level of knowledge gaps. When the uncertainty exceeds certain threshold, the authors propose to construct a tree of knowledge gap, where each child node is a valid subset of knowledge space. The branching is controlled via LLM, with pruning and generate human query. Human will answer the question to narrow the space of knowledge gap and the search is stopped when the action uncertainty is acceptable. Through experiment, demosntrate strong performance over baseline.

**Strengths:**

* Strong Motivation: incomplete information, long-standing challenge

* Principled Elicitation Trigger: The strength lies in the formalization of when to ask for human help. Using the Q-value variance over the knowledge gap ($\sigma_u^2(s)$) is a sound way to quantify the policy's uncertainty. It allows the agent to trigger human interaction only when the uncertainty is high enough to impact decision-making, rather than querying at every step or based on simple heuristics.

* Novel Neuro-Symbolic Combination: The idea of combining a neural policy (for value estimation) with a symbolic tree (for knowledge representation) is interesting. Using an LLM to bridge the gap between this symbolic tree and a natural language query for the human is a novel pipeline.

**Weaknesses:**

* Lack of Baselines: The baselines of Pure LLM fails to serve as a planner, which totally makes sense. The baseline of "Query-A" queries for an expert action but not for information, which can be viewed as an ablation of MINT module. The core of MINT is its ability to actively elicit information to resolve "knowledge gaps". Therefore, the evaluation is missing a crucial comparison to the most relevant body of work: agents that actively seek information (not expert actions) to resolve model uncertainty, particularly regarding unknown rewards or transitions. Works on active preference elicitation (e.g., Handa et al., 2024) and active queries in RLHF (e.g., Ji et al., 2024) represent a more appropriate set of baselines. Without these, it is difficult to assess the novelty and efficiency of MINT's specific query-generation strategy.

  * Handa, Kunal, et al. "Bayesian preference elicitation with language models." arXiv preprint arXiv:2403.05534 (2024).
  * Ji, Kaixuan, Jiafan He, and Quanquan Gu. "Reinforcement learning from human feedback with active queries." arXiv preprint arXiv:2402.09401 (2024).

* Simulated Human: For a paper on "human-AI teaming," the experimental setup excludes the human entirely. The paper states, "In our experiments, we use another LLM with full knowledge to automatically generate the yes/no answers" (line 321). This totally makes sense for simple cases, but requires more emphasis, as it invalidates many of the core challenges of this field (human vagueness, incorrect answers, cognitive load, query phrasing) are all ignored.

* The "Grounding" Problem: The paper misses a critical link in its pipeline and experiment setup: grounding a sensory input (e.g., "raw pixels" in Atari) to a symbolic knowledge gap $u$. The MINT process is supposedly triggered by "an uncertain object detected by the perception module" (line 240), which implies a separate, unexplained perception system. This system must identify the sole source of the policy's high variance and map it to the abstract root node $u_0 = \langle \text{any, any, 0, 1} \rangle$.

**Questions:**

* The Grounding Problem: Following Weakness, can the authors please explain the mechanism that connects a specific sensory input (e.g., a "blue block" in MiniGrid) to the triggering of the MINT process? Specifically,how does the system: (1) map high Q-value variance back to a specific sensory input (2) instantiate the abstract root node $u_0$ to correspond to that specific object? How does the system handle cases with multiple unknown objects, and how is variance attributed?

* Definition of $u$: Following the grounding problem, how is the space of $u$ cosntructed, is it predefined for each experiment? The paper's expansion logic (line 294-305) seems to assume these dimensions are orthogonal. What is the justification for this assumption? How would MINT handle a single unknown object where, for example, the impact on reward and transition is intercorrelated?

* Lack of Experiment Details: what is the convergence of bootstrapped DQN, as it learns across a wide distribution of different knowledge gap?

---

> ### Author Response · Authors · 2025-11-20
>
> > Weakness: Lack of Baselines
>
> We agree that active information-seeking methods, including Bayesian preference elicitation with LMs and RLHF with active queries, are highly relevant conceptually. Our work, however, addresses a different operational setting:
>
> •	(1) optimizes natural language preference queries in a static preference space to learn a reward model or user utility, but does not reason about environment dynamics or object-level uncertainty in an MDP.
>
> •	(2) designs query-efficient algorithms for RLHF, where active queries decide which preference labels to solicit to better align a parametric policy, again without explicit object-centric reasoning about unknown transitions/rewards in an MDP.
>
> In contrast, MINT:
>
> •	(1) Models knowledge gaps over environment dynamics and rewards via a descriptor space Φᵤ that parameterizes transition kernels and rewards
>
> •	(2) Builds a neuro-symbolic tree over these gaps and uses uncertainty-aware Q-values to reason about how each potential query reduces the return gap of a downstream planner
>
> •	(3) Provides a pseudo-Lipschitz bound on the return loss in the extended MDP family, which is not present in the above active-preference works.
>
> That being said, in the revision, we add a “LLM + Query” baseline that, upon high Q-variance around an uncertain object, LLM automatically queries a human directly for questions directly generated by itself, without building a tree. Please check our response to reviewer McDF. This is an active-information baseline that uses the same UA-DQN and descriptor parameterization as MINT but lacks symbolic reasoning. We believe these additions will make it clearer that MINT’s advantage comes from structured knowledge-gap reasoning and query optimization rather than simply “asking more questions.”
>
> > Weakness: Simulated Human
>
> MINT always trusts human responses as expert knowledge, following the settings in prior works [1,2]. Nevertheless, an ambiguous answer other than yes/no will be treated as a signal for no available new knowledge in that specific field, i.e., $u_{k+1}=u_k$. In that case, MINT will either try generating another query asking for different information or default to its current best action under partial observation using the neural network, depending on the current tree structure and query limit.
>
> Also, an incorrect or noisy answer can mislead an AI agent and not necessarily reduce the knowledge gap in planning, causing poor performance. To this end (and inspired by your great questions), we conducted additional experiments by introducing intentionally misleading responses into our human-agent interaction scenarios. Please check the results from our rebuttal to reviewer McDF. The results are trivial since the mistakes made by humans are still trusted, and mislead the MINT agent to fail.
>
> Besides, to include data from true humans, an IRB review for human involvement in research is required, and we are now in the process of securing that.
>
> [1] Li, Boning, and Longbo Huang. "Efficient Online Pruning and Abstraction for Imperfect Information Extensive-Form Games." The Thirteenth International Conference on Learning Representations.
>
> [2] Li, Boning, Zhixuan Fang, and Longbo Huang. "RL-CFR: improving action abstraction for imperfect information extensive-form games with reinforcement learning." Proceedings of the 41st International Conference on Machine Learning. 2024.
>
> > Question: Lack of Experiment Details
>
> In the revision, we will add training curves (return vs environment steps) for UA-DQN in MiniGrid and Atari, aggregated over 5 seeds, to show stable convergence despite the wider descriptor distribution.

---

> ### Author Response · Authors · 2025-11-20
>
> > Weakness and Questions: The "Grounding" Problem
>
> > Questions: Definition of $u$
>
> In the current experiments, the “perception module” is deliberately simple and environment-specific:
>
> MiniGrid: We explicitly insert one or more “uncertain blocks” as distinct object types in the grid. The simulator annotates their positions and that they are unknown.
>
> Atari Pacman: We similarly insert colored rectangular regions with unknown reward or stochastic terminal behavior.
>
> Isaac Search-and-Rescue: The simulator tags low-confidence objects (e.g., smoke-obscured boxes, potential victims) using metadata from the 3D reconstruction and detection pipeline.
>
> In all cases, the presence of an uncertain object is provided symbolically by the environment itself, and this is precisely what we mean by “detected by the perception module.” The current paper does not claim to solve the full vision problem of grounding from raw pixels to objects; rather, it focuses on what to ask once a candidate unknown object has been flagged.
>
> We will clarify this in Sec. 4.1/4.2 and in the experiment descriptions.
> (1) Mapping Q-variance back to a specific sensory input/object.
> Given a state s and a set of uncertain objects $O = \{o_1,…,o_n\}$, our UA-DQN takes as input $(s, \Phi_u)$, where the descriptor space $\Phi_u$ encodes all current per-object gaps. During training, we generate experience both with the full descriptor $\phi ^ * $ and with masked versions corresponding to different knowledge gaps $u$ (Algorithm 1).
> At deployment, we use the known structure of $\Phi_u$: each uncertain object contributes its own component (type_i, subtype_i, $w_i \in [0,1]$) to the descriptor. When an “uncertain object” is flagged in the state, we associate one root gap u₀ per object with initial $\Phi_u$ = ⟨type=any, subtype=any, $w\in[0,1]$⟩ for that object, while the others are held at their current best-known values. For that object-specific $u$ we compute $\sigma _ u ^ 2 (s, a ^ *)$ from UA-DQN; if it exceeds $\delta$, we trigger MINT for that object.
> Thus, the mapping from high variance to object is not done via saliency over raw pixels, but through the object-indexed descriptor components we control in the simulator.
>
> (2) Instantiating $u_0$ and handling multiple unknown objects.
> Root node $u_0$. As described in Sec. 4.2, each MINT node stores u as a tuple ⟨type, subtype, $w_{min}$, $w_{max}$⟩ summarizing the agent’s belief about one unknown object. The root $u_0$ = ⟨any, any, 0,1⟩ simply means “we do not yet know whether this object affects rewards or transitions, nor whether its effect is positive/negative or deterministic/stochastic, and its magnitude lies in [0,1].”
> Multiple unknown objects. In MiniGrid and Atari experiments, when there are 3 or 5 unknown objects (Table 1), we instantiate a separate MINT tree for each object and resolve them sequentially (e.g., in decreasing order of estimated impact on $\sigma_u (s, a^*)$). This matches the environment design where each uncertain block affects a disjoint set of positions or rewards; i.e., their contributions are additive and do not interact in the transition kernel. We will add this explanation and a figure in the appendix.
>
> (3) Construction of the space of $u$ and orthogonality of dimensions.
> For each environment, $\Phi_u$ is indeed predefined but general:
> type ∈ {reward, transition}
> subtype is a small discrete set, e.g., {positive reward / negative penalty / terminal stochasticity}.
> $w \in [0,1]$ bounds either reward magnitude or transition failure probability.
> The expansion rule (type → subtype → w interval splits) is a search heuristic, not a theoretical assumption of orthogonality. Our main theorem only assumes that $\Phi_u$ is bounded and that we can bound the MDP distance between extreme descriptors $(\Phi_1, \Phi_2)$; it does not require independent dimensions.
> In the current experiments, we restrict each unknown object to affect either transitions or rewards (but not both), which respects the environment construction. We will add this as an explicit assumption and discuss how, in more general settings, φ could be higher-dimensional to encode jointly correlated transition/reward changes, with MINT operating over that joint descriptor.

---

### Official Review · Reviewer_McDF · 2025-10-31

**Soundness:** 1
**Presentation:** 2
**Contribution:** 3
**Rating:** 2
**Confidence:** 2

**Summary:**

The paper proposes MINT, which is a method for human-in-the-loop planning in environments with unknowns and incomplete information. The method aims to discover strategies for optimally eliciting human inputs to aid in planning. It learns a neural planning policy to estimate uncertainty and uses an LLM to create queries that elicit human input to optimize performance. The method performs very well in a set of benchmark environments with unseen and unknown objects with a limited number of human queries.

**Strengths:**

- The paper frames the problem of human-in-the-loop very clearly. It addresses the practical challenge of determining what to ask and when to ask in human-AI planning, and solves this practical challenge in a novel way.
- The formalization with a knowledge gap and a descriptor space is elegant and makes a lot of sense. Comparing the estimated Q-gap to the variance at the node to determine when to query for more information is a nice idea.
- Using an LLM to craft binary questions that maximize information gain is a sound way to involve an LLM in the planning procedure.
- The results on the Isaac Gym benchmark are impressive, and the proposed MINT* clearly outperforms the LLM baselines used.

**Weaknesses:**

- Could you please clarify the inequality $\max_a \Delta_{s_0,a} (M_{\phi*}, M_{\phi_1}) \le \max_a \Delta_{s_0,a} (M_{\phi_1}, M_{\phi_2})$ in the proof of Theorem A.4?  For instance, in the case of $\gamma = 0$, this seems only to hold if the interpolation $\phi^* = \lambda\phi_1 + (1-\lambda)\phi_2$ also implies an interpolation in the MDP space (or reward function, to be precise). Wouldn't you need additional assumptions about the behavior of the reward functions to make this claim in general? For $\gamma > 0$, the symmetrization of two asymmetric divergences with the minimum does not, in general, satisfy the triangle inequality, which could be an additional problem.
- The discussion surrounding the aleatoric and epistemic uncertainty could be clarified. Typically, aleatoric uncertainty refers to uncertainty that cannot be reduced by collecting more data. Hence, when we ask for information from humans, we're trying to reduce the epistemic uncertainty. However, the authors state (L259) that the paper focuses on aleatoric uncertainty. Then, the aleatoric uncertainty estimation from UA-DQN is used (L267). However, $\phi$ (or $\Phi_u$) is used as input to $Q$. Hence, the aleatoric estimate from UA-DQN might actually incorporate variance from the uncertainty associated with the knowledge gap and some model-weight variation, i.e., it correlates with epistemic uncertainty, which most likely enables the method to work. If you examine Table 6 in the appendix, the uncertainty declines as you ask questions, which, to me, seems to be epistemic uncertainty reduction by definition.
- The method assumes full observability of the environment during training, which gives it an advantage over the baseline RL algorithms. On the other hand, MINT relies on RL training to outperform the zero-shot LLM baselines. Table 4 reveals that this method uses 5e7 training episodes, which could make the comparison unfair. Even then, this training only allows us to solve one specific domain. The paper would benefit from an LLM baseline that can either query a human or act, to help understand the benefits of the explicit planning and the UA-DQN formulation. To make his baseline more competitive, it could even be RL-trained (think GRPO or some variant). However, this RL training is likely outside the scope of the rebuttal and would be a valuable contribution on its own.
- While the MINT* algorithm used for the NVIDIA Isaac task arguably fixes the RL-related weakness of MINT, it is severely underdescribed in the paper. It feels like a conceptually very big leap from the UA-DQN algorithm that the paper discusses.
- The method assumes that the different types of uncertainty are pre-specified, whereas an LLM could perhaps detect uncertainty on the fly. Furthermore, the method assumes that humans always give correct and complete answers when queried.
- There's related work the authors might want to acknowledge. Even though they don't involve direct human interaction, they let agents query external sources. [1] learns a cost-aware policy for when to query an LLM for high-level instructions. Additionally, there's work on LLMs that query for additional information to improve reasoning and decision-making [2, 3].

[1] Hu, B., Zhao, C., Zhang, P., Zhou, Z., Yang, Y., Xu, Z., & Liu, B. (2023). Enabling intelligent interactions between an agent and an LLM: A reinforcement learning approach. RLC.

[2] Kim, M., Kim, J., Kim, J., & Hwang, S. W. (2024, November). QuBE: Question-based belief enhancement for agentic LLM reasoning. In Proceedings of the 2024 Conference on Empirical Methods in Natural Language Processing (pp. 21403-21423).

[3] Chen, X., Zhang, S., Zhang, P., Zhao, L., & Chen, J. (2023). Asking before acting: Gather information in embodied decision making with language models. arXiv preprint arXiv:2305.15695.

**Questions:**

See above. In general, the method is exciting, and the problem being tackled is very important. However, the authors should clarify the assumptions underlying the proof of Theorem A.4 and the discussion surrounding the types of uncertainty. They should also consider including a naive LLM baseline with the capability to query for information and properly describe the MINT* algorithm. If the authors can clarify/address these issues, I'm willing to raise my score.

---

> ### Author Response · Authors · 2025-11-20
>
> > Weakness: Could you please clarify the inequality ...
>
> Thanks for pointing this out. Yes, there is a Lipschitz continuity of the optimal Q function with respect to the underlying MDP (a set of MDPs corresponding to the knowledge gap in our case).  This Lipschitz continuity has been well studied in the lifelong reinforcement learning literature, such as [1]. We revised the proof of Theorem A.4 and corrected some minor issues or confusing parts in the proof, which will not compromise the foundation of our theory, especially the justification of bounding the overall return on the true uncertainty by iteratively limiting the uncertainty range.
>
> Here, we provide a brief clarification on the inequality correction:
> With $\phi ^ * = \lambda \phi_1 + (1-\lambda) \phi_2$, we refer to a new MDP $M_{\phi ^ *}$ with $T_{\phi ^ *} = \lambda T_{\phi _ 1} + (1 - \lambda) T_{\phi _ 2}$ and $R_{\phi ^ *} = \lambda R_{\phi _ 1} + (1 - \lambda) R_{\phi _ 2}$
>
> For Theorem A.4 specifically, we admit that the original inequality still needs elaborate assumptions on the value function for proof. But to keep the theorem sound, we can replace the previous local dissimilarity $\Delta_{s,a}$ with the global dissimilarity $\Delta(M_1, M_2) = \frac{1}{1 - \gamma} \max_{s, a}( |R_1(s,a) - R_2(s,a)| + \frac{ \gamma R_{max}}{1 - \gamma} \sum _ {s'}| T_1 (s' | s, a) - T_2 (s' | s, a)| )$.
> As the upper bound of $\Delta_{s,a}$, the global dissimilarity still satisfy all the theorem about $\Delta_{s,a}$, while it's easy to prove that $\Delta(M_{\phi _ *}, M_{\phi_1}) \leq  \Delta(M_{\phi _ 1}, M_{\phi_2})$.
>
> [1] Zhang, Zuyuan, and Tian Lan. "Lipschitz Lifelong Monte Carlo Tree Search for Mastering Non-Stationary Tasks." arXiv preprint arXiv:2502.00633 (2025).
>
> > Weakness: The discussion surrounding the aleatoric and epistemic uncertainty could be clarified...
>
> We feel that this is more like aleatoric uncertainty instead of epistemic uncertainty. It may depend on how you view the questions/answers – which we consider as improved belief to further delineate the MDP, rather than additional training data/observations that are often associated with epistemic uncertainty. The aleatoric uncertainty refers to uncertainty that cannot be reduced by **collecting more interactive data via model exploration**, i.e., the uncertainty is inherent in the environment. That is the case because if you take this view, the model is trained across a range of distributions of different knowledge gaps (i.e., different MDPs with distinct transition and reward functions), even though some of them share the same observation (uncertainty descriptor). Therefore, from this perspective, this type of uncertainty cannot be reduced with more training data/observations. Instead, it’s determined by the distributions of MDPs conditioned on the uncertainty descriptor, and the uncertainty reduces when we have a better belief of the MDP due to the questions/answers. We believe that the use of aleatoric uncertainty is appropriate in this case.
>
> But we completely see your point and where the confusion comes from. We will carefully revise this part of the discussion and clarify the types of uncertainty.
>
> > The method assumes full observability of the environment during training ...
>
> Thanks for mentioning these, but we argue that there are some misunderstandings. First, during training, the environment is NOT fully observable. Instead, agents with any RL algorithm are given training data with both a randomly masked uncertainty descriptor and a ground truth descriptor. Therefore, there is no such “advantage” over the baseline RL algorithms. Secondly, these LLM models are already well-trained with a huge amount of data and, in most cases, call existing well-performing path-planning algorithms as tools to solve the problem (in MiniGrid), while the RL agent bootstrapped itself through training. Therefore, we don’t agree that the comparison is “unfair”.
>
> That being said, we provide an additional LLM baseline (GPT-4o) in MiniGrid and Isaac environment that enables the LLM to ask limited questions before acting, but without MINT framework:
>
> | MiniGrid | 1-uncertainty success | 1-uncertainty Avg. Rew. | 3-uncertainty success | 3-uncertainty Avg. Rew. | 5-uncertainty success | 5-uncertainty Avg. Rew. |
> | - | - | - | - | - | - | - |
> | LLM | 100 | 9.26±0.42 | 99 | 9.30±1.36 | 96 | 8.75±2.72 | 95 | 8.61±2.92 |
> | LLM + Query | 100 | 9.49±0.89 | 100 | 9.55±1.93 | 97 | 9.02±2.33 |
> | MINT Limited | 99.6±0.5 | 9.29±1.47 | 100±0.0 | 9.90±1.09 | 97.0±0.9 | 9.56±2.31 |
>
> From the result, we see that enabling the query did improve the success rate compared with pure LLM since it now identifies risky uncertainties. However, the average reward is still lower than MINT due to the elicitation strategy.
> The RL-training LLM baseline mentioned by the reviewer is out of the scope of this work, but it is an interesting future direction.

---

> ### Author Response · Authors · 2025-11-20
>
> > Weakness: While the MINT* algorithm used for the NVIDIA Isaac task arguably fixes the RL-related weakness of MINT...
>
> MINT* is the same algorithm, but a generalization of the same MINT framework on a more realistic, continuous-control environment, such as the Isaac environment. The core MINT components remain exactly the same: We still build a knowledge-gap tree whose nodes correspond to subsets of the uncertainty/knowledge gap, and whose edges correspond to candidate yes/no queries. We use the same algorithm to evaluate each node by the variance of rewards and differences in actions to decide which branch to expand and which queries to ask. While the task environment certainly becomes 3D high-fidelity simulation in this case, the same algorithm for knowledge gap reasoning applies, with an underlying planner that works in 3D. Due to the page limit, we could not provide full details in the paper.
>
> The only modification in MINT* is the underlying planner:
> In MiniGrid and Atari, this planner is UA-DQN (discrete actions, grid-like state). In the Isaac Search-and-Rescue scenario, we must handle continuous 3D control and high-dimensional observations, where UA-DQN is not appropriate. MINT* therefore utilizes LLM and existing path-planning tools, such as classical path-planning algorithms like rapidly exploring random tree, to directly plan the path after the uncertainty has been resolved. Here, the LLM will analyze the environment and decide the subgoals and the possibility of uncertainties as a higher-level planner, while passing these commands and planning paths through the lower-level planner.
>
> We will include and clarify these details in the revisions. We hope that this clarification makes clear that MINT* is a straightforward instantiation of the same neuro-symbolic framework in a more realistic continuous-control setting, rather than a separate, under-described algorithm.
>
> > The method assumes that the different types of uncertainty are pre-specified ...
>
> There are two parts to this question. First, it is great you point out that there are different types of uncertainty in an open world environment. But our paper focuses on planning with respect to uncertainty in the MDP context (which is applicable many practical problems). For this purpose, the way different types of uncertainty can affect planning is limited in the sense that they mainly affect (i) state transition probabilities and (ii) reward function. MINT focuses on these potential impacts of uncertainty on MDP-based planning. Yes, LLM could detect new types of uncertainty on the fly, but their impacts to our planning problem are mainly through the same mechanism. Further, we would like to mention that the same design of MINT can be applied to different types of uncertainty we may encounter an open world environment. Of course, it may require building new reasoning trees and obtaining new questions using LLM. Enabling such real-time interactions and reasoning is a research topic on its own. We will further discuss this in the future work.
>
> We are also aware that human responses can introduce another source of uncertainty, besides the planning uncertainty we consider in this paper, as the responses can be noisy, inaccurate, or even completely false. It could mislead an AI agent and not necessarily reduce the knowledge gap in planning. MINT considers trusted human responses as expert knowledge, following the settings in prior works [1,2].
> It is our plan to consider the uncertainty of human responses in our future work. There are a few ways to address the problem: (1) We can introduce an uncertainty model of human responses, e.g., a belief model following Bayesian inference to quantify the uncertainty and factor the distribution over $\Phi_u$ into planning, like [3]. (2) We can leverage recent results in trust to reason and develop a new module to establish human-AI trust in planning, e.g., by discounting different branches, etc.
>
> Also, we conduct an additional evaluation on this problem when humans give random answers:
>
> | Method |MiniGrid (3 objects): success% | avg. rew. | Atari |	Isaac: target 1	| target 2|
> | - | - | - | - | - | - |
> | MINT | 100 | $9.90 \pm 1.09$ | $411.1\pm 28.8$ | 100 | 95 |
> | 50% Random | 95 | $ 8.21 \pm 4.83$ | $373.2 \pm 35.1$ | 72 | 74 |
> | 100% Random| 91 | $ 7.81 \pm 5.79$ | $358.4 \pm 51.2$ | 61 | 52 |
>
> The performance drops as the random rate increases, which is not surprising since MINT always trusts humans.
>
> [1] Da Silva, Felipe Leno, et al. "Uncertainty-aware action advising for deep reinforcement learning agents."
>
> [2] Singi, Siddharth, et al. "Decision making for human-in-the-loop robotic agents via uncertainty-aware reinforcement learning."
>
> [3] Trick, Susanne, et al. "Interactive reinforcement learning with Bayesian fusion of multimodal advice."
>
> > Weakness: There's related work the authors might want to acknowledge...
>
> Thanks for providing these related works, and we will add a discussion of them in the revision

---

### Author Response · Authors · 2025-11-20

Dear Area Chair,

Thank you sincerely for overseeing the evaluation of our submission under the unusual situation involving a reviewer leak and reassignment. We appreciate your efforts to ensure a fair process. While the discussion phase saw limited engagement from reviewers, all reviews rated the paper with good contribution.

Our work introduces MINT, which is a Minimal Information Neuro-Symbolic Tree to address the uncertainty by actively eliciting human inputs via LLMs in object-driven planning tasks. This process provides a two-way communication between the human and the agent to narrow down the knowledge gap with a minimal number of queries. Compared to previous works which mostly ask for expert actions, our method addresses the uncertainty with additional information via human-AI interaction. We also provided empirical results and theoretical proofs to further reinforce the soundness of our method.

During the review phase, we have provided additional experiments and clarifications that directly addressed key concerns:

(1)	Vague/Inaccurate human responses (McDF, 6Xak, fY71): MINT always trusts human responses as expert knowledge, following the settings in prior works in which an expert action is given after a query. It is our plan to consider the uncertainty of human responses in our future work, and we briefly discussed some potential measures for that. Also, we conduct an additional evaluation on this problem when humans give random answers.

(2)	 “LLM + Query” baseline (McDF, 6Xak): We added another baseline in which LLM automatically queries a human for questions directly generated by itself upon high Q-variance around an uncertain object, without building a tree.

(3)	The grounding/observation problem (McDF, 6Xak, fY71): The current paper does not claim to solve the full vision problem of grounding from raw pixels to objects; rather, it focuses on what to ask once a candidate unknown object has been flagged. We clarified how the knowledge gap descriptor is constructed and how the uncertainty object is perceived in detail.


Besides, we briefly clarified some misunderstandings of our paper raised by the reviewers:

(1)	Uncertainty type: We argue that the model aims to estimate the aleatoric uncertainty instead of epistemic uncertainty. That is the case because the model is trained across a range of distributions of different knowledge gaps, even though some of them share the same observation (i.e., knowledge gap descriptor). From this perspective, this type of uncertainty cannot be reduced with more training data/observations. Furthermore, we narrowed the way different types of uncertainty can affect planning to (i) state transition probabilities and (ii) reward function. MINT focuses on these potential impacts of uncertainty on MDP-based planning.

(2)	Issue in theoretical proof: We corrected some minor issues or confusing parts to provide a more solid proof, which will not compromise the foundation of our theory. We also pointed out that Lipschitz continuity has been well studied in lifelong reinforcement learning and provided some related works.

(3)	MINT* algorithm used for the NVIDIA Isaac task: We clarified the details of MINT*, which is a generalization of the same MINT framework on a more realistic, continuous-control environment.

Overall, our work makes a novel and solid contribution, with extensive empirical results already demonstrated in the original submission. The additional clarifications and experiments provided during the review process further reinforce its soundness. We respectfully appreciate the AC’s consideration during the meta-review and hope our work’s impact and value to the ICLR community will be recognized.

---

### Meta-Review · Area_Chair_qQiY · 2026-01-07

**Summary:**

1) A central concern across reviews (although not explicit) is that the paper’s core contribution is not sharply defined. The submission combines several components (uncertainty-aware RL, a hand-specified knowledge-gap descriptor, symbolic tree search, and LLM-generated yes/no questions) into a working pipeline, but it remains unclear whether the main claim is algorithmic, conceptual, or systems-oriented. Reviewers expressed difficulty isolating a general, reusable insight beyond the observation that structured querying helps when object-level uncertainty is already exposed by the environment.

Supporting reviewer quotes:
- Reviewer McDF: “It feels like a conceptually very big leap from the UA-DQN algorithm that the paper discusses.”
- Reviewer 6Xak: “Without these [baselines], it is difficult to assess the novelty and efficiency of MINT’s specific query-generation strategy.”


2) Reviewers raised concerns about generality and realism. Key assumptions underlying the empirical success, such as simulator-provided identification of uncertain objects, hand-crafted gap parameterizations, and trusted binary human responses, limit the extent to which conclusions can be transferred beyond the designed benchmarks. This led to questions about whether the approach would scale to settings with learned or compositional uncertainty, realistic perception, or noisy human interaction.

Supporting reviewer quotes:

- Reviewer 6Xak: “The experimental setup excludes the human entirely.”
- Reviewer 6Xak: “Many of the core challenges of this field (human vagueness, incorrect answers, cognitive load, query phrasing) are all ignored.”
- Reviewer fY71: “It is unclear how to scale the type-subtype-value scheme when unknowns are high dimensional or compositional.”

3) While the empirical results are strong in the chosen environments, reviewers noted that the evaluation does not fully substantiate the broader claims about human-in-the-loop planning. In particular, the reliance on simulated human answers, the mismatch between heavily trained RL components and largely zero-shot LLM baselines.

Supporting reviewer quotes:

- Reviewer McDF: “This method uses 5e7 training episodes, which could make the comparison unfair.”
- Reviewer 6Xak: “The paper misses a critical link in its pipeline… grounding a sensory input to a symbolic knowledge gap.”

4) A concern is the paper’s treatment of aleatoric versus epistemic uncertainty. Although the authors frame knowledge-gap uncertainty as aleatoric, the core interaction mechanism (eliciting information and updating the feasible descriptor set) closely resembles epistemic uncertainty reduction in the standard sense, leading to conceptual ambiguity. Importantly, the theoretical analysis does not depend on this distinction: the return bounds rely solely on shrinking the set of possible MDPs, regardless of the uncertainty’s source. Consequently, the aleatoric/epistemic framing functions mainly as a post hoc justification for the chosen uncertainty estimator rather than as a substantive element that clarifies or strengthens the theoretical contribution.

Supporting reviewer quotes:

- Reviewer McDF: “The discussion surrounding the aleatoric and epistemic uncertainty could be clarified.”

**Reviewer Concerns:**

1) The rebuttal clarifies the scope of the contribution and explains the intended positioning of the work, but it does not substantially resolve the concern about the absence of a single, crisp core finding. The additions and explanations improve understanding of how the system works, yet the paper still reads as a collection of coordinated design choices rather than a sharply articulated methodological or conceptual insight.

2) The rebuttal explicitly acknowledges limitations around grounding, hand-designed descriptors, and idealized human responses, and appropriately scopes these issues as outside the paper’s focus or as future work. However, these assumptions remain fundamental to the method’s effectiveness, and thus the concern about limited generality and reliance on engineered interfaces is only partially addressed.

3) The authors add an “LLM + Query” baseline and provide additional experiments and theoretical clarifications, which address several reviewer questions about fairness and soundness. Nonetheless, the evaluation still does not convincingly disentangle which components drive performance or demonstrate robustness to realistic human behavior,

4) The rebuttal clarifies the authors’ intended interpretation of knowledge-gap uncertainty as aleatoric and acknowledges potential confusion in terminology. However, the core concern remains unresolved: the query-and-update mechanism functions as belief refinement over possible MDPs and thus closely resembles epistemic uncertainty reduction, and the theoretical analysis does not rely on this distinction. As a result, the aleatoric/epistemic framing remains conceptually ambiguous and does not play a substantive role in the method or its theoretical justification. Furthermore, the theoretical contribution seems to be itself a bit out-of-place: it does not depend in any way on the proposed algorithm. The main part of Theorem 4.4 seems to be a rather generic interpolation argument and largely based on the work of Lecarpentier et al., 2021), as cited by the authors in the appendix. Furthermore, it is not clear what the theory buys us, as it does not give us any query strategy.

**Reviewer Scores:**

Hard to tell, there was absolutely no review engagement during the rebuttal phase it could be that Reviewer McDF would go 2 -> 4

---

### Decision · Program_Chairs · 2026-01-26

Reject